# Constraining ecosystem model with Adaptive Metropolis algorithm using boreal forest site eddy covariance measurements

Jarmo Mäkelä[1], Jouni Susiluoto[1], Tiina Markkanen[1], Mika Aurela[1], Heikki Järvinen[2], Ivan Mammarella[2], Stefan Hagemann[3], and Tuula Aalto[1]

[1]Finnish Meteorological Institute, P.O. Box 503, 00101 Helsinki, Finland
[2]Department of Physics, P.O. Box 48, 00014 University of Helsinki, Finland
[3]Max Planck Institute for Meteorology, Bundesstraße 53, 20146 Hamburg, Germany

*Correspondence to:* Jarmo Mäkelä (jarmo.makela@fmi.fi)

**Abstract.** We examined parameter optimization in JSBACH ecosystem model, applied for two boreal forest sites (Hyytiälä and Sodankylä) in Finland. We identified and tested key parameters in soil hydrology and forest water and carbon exchange related formulations and optimized them using the Adaptive Metropolis algorithm (AM) for Hyytiälä with a five year calibration period (2000–2004) followed by a four year validation period (2005–2008). Sodankylä acted as an independent validation site, where optimizations were not made.

The tuning provided estimates for full distribution of possible parameters, along with information about correlation, sensitivity and identifiability. Some parameters were correlated with each other due to phenomenological connection between carbon uptake and water stress or other connections due to the set-up of the model formulations. The latter holds especially for vegetation phenology parameters. The least identifiable parameters include phenology parameters, parameters connecting relative humidity and soil dryness, and the field capacity of the skin reservoir. These soil parameters were masked by the large contribution from vegetation transpiration.

In addition to leaf area index and maximum carboxylation rate, the most effective parameters adjusting the gross primary production (GPP) and evapotranspiration (ET) fluxes in seasonal tuning were related to soil wilting point, drainage and moisture stress imposed on vegetation. For daily and half-hourly tunings the most important parameters were the ratio of leaf internal $CO_2$ concentration to external $CO_2$ and the parameter connecting relative humidity and soil dryness. Effectively the seasonal tuning transferred water from soil moisture into ET, and daily and half-hourly tunings reversed this process.

The seasonal tuning improved the month-to-month development of GPP and ET, and produced the most stable estimates of water use efficiency. When compared to the seasonal tuning, the daily tuning is worse on the seasonal scale. However, daily parametrization reproduced the observations for average diurnal cycle best, except the GPP for Sodankylä validation period, where half-hourly tuned parameters were better. In general, the daily tuning providing the most reduction in model-data mismatch.

The models response to drought was unaffected by our parametrizations and further studies are needed into enhancing the dry response in JSBACH.

# 1 Introduction

Inverse modelling of ecosystem model parameters against in situ observations is an established way to tune model parameters (e.g. Scharnagl et al., 2011). As observation sites have their own characteristics, it is necessary to make local site simulations for model evaluation and calibration as they may reveal new insight into model behaviour and guide further development.

Model-data fusion has been applied for boreal forest sites by e.g. Aalto et al. (2004); Peltoniemi et al. (2015b); Thum et al. (2007, 2008); Wu et al. (2011).

In this study we perform site level parameter optimization in the JSBACH model (Kaminski et al., 2013; Knorr and Kattge, 2005; Reick et al., 2013). JSBACH is the land surface component of the Earth System model of Max Planck Institute for Meteorology (MPI-ESM), used to simulate water and carbon storages and fluxes in the soil-vegetation-atmosphere continuum.

The water and carbon fluxes are coupled in the model and thus modification of parameters related to one component usually has an effect on the others as well. The optimization process and the optimized values are also affected by the assimilation frequency and interval in minimizing the model-data mismatch. This effect can be studied in numerous ways e.g. Santaren et al. (2014) varied the length of assimilation interval while Matheny et al. (2014) focused on the diurnal error patterns.

The motivation for this study comes from results showing that CMIP5 model simulations, one of which is MPI-ESM, have

15 systematic evapotranspiration biases over continental areas (Mueller and Seneviratne, 2014). These kinds of biases have significant implications for climate change projections (Boé and Terray, 2008) but also have distinctive behaviour on a regional scale. In addition a comparative study of the gross primary production (GPP) of Finnish forests (Peltoniemi et al., 2015a) revealed that JSBACH has an insufficient response to water limitation in Finland – it tends to overestimate GPP and evapotranspiration during dry periods. This is especially apparent in the dry year 2006 as JSBACH is unable to transfer the reduced rainfall into

20 lower levels of GPP.

In this study we apply the JSBACH ecosystem model for Hyytiälä (Kolari et al., 2009; Suni et al., 2003) and Sodankylä (Aurela, 2005; Thum et al., 2008) sites. We identify key parameters in soil hydrology and evapotranspiration related formulations and test their effectiveness with elementary methods. We study the effect of different timescales (seasonal, daily and half-hourly) on the assimilation process and the effect of this on the optimized parameter values. Several optimizations are

25 performed using the Adaptive Metropolis (AM) algorithm over a five year calibration period (2000–2004) followed by a four year validation period (2005–2008).

The goals of this study are to test the applicability of the AM optimization method for JSBACH and the impact of different temporal resolutions on the optimization process, and to improve the models response to environmental drivers, focusing on dryness.

## 2 Materials and methods

### 2.1 Measurements, sites and instrumentation

In this study we use site level data from two Finnish measurement sites: Hyytiälä (61°51'N, 24°17'E, 180 m a.s.l.) and Sodankylä (67°22'N, 26°38'E, 179 m a.s.l.). These well-established sites have long continuous measurement data sets representing well the southern and northern boreal Finnish coniferous evergreen forests. The data used in this study is available for the scientific community through various databases such as FLUXNET (doi:10.17616/R36K9X).

Hyytiälä site is a Finnish Scots pine (*Pinus sylvestris*) forest (Kolari et al., 2009), planted in 1962 after burning and mechanical soil preparation. The soil type in Hyytiälä is *Haplic Podzol* on glacial till and the site is of medium fertility (Kolari et al., 2009). The forest also has sparse understory of Norway Spruce (*Picea abies*) and scattered deciduous trees. The maximum of measured all-sided leaf area index (LAI) is 6.5 $m^2$ $m^{-2}$ for the Scots pine. The carbon dioxide ($CO_2$) and water vapour ($H_2O$) fluxes between vegetation and atmosphere have been measured in Hyytiälä continuously since 1997 (Vesala et al., 2005).

The Sodankylä forest, in Sodankylä at the Finnish Meteorological Institute's Arctic Research Centre, is also a Scots pine forest (*Pinus sylvestris*) with maximum measured LAI of 3.6 $m^2$ $m^{-2}$ as determined from a forest inventory in early autumn (Thum et al., 2007). The forest on *Fluvial Sandy Podzol* has been naturally regenerated after forest fires with tree age ranging approximately from 50 to 100 years. The sparse ground vegetation consists of lichens (73%), mosses (12%) and ericaceous shrubs (15%). The $CO_2$ and $H_2O$ flux measurements in Sodankylä have been running since 2000 (Aurela, 2005).

The $CO_2$ and $H_2O$ fluxes were measured by the micrometeorological eddy covariance (EC) method which provides a direct measurement of the mass and energy exchange between the atmosphere and the biosphere averaged on an ecosystem scale. In the EC method, the flux is obtained as the covariance of the high frequency (10 Hz) observations of vertical wind speed and the constituent in question (Baldocchi, 2003). The $CO_2$ fluxes were corrected for the storage change below the measurement height to accurately estimate the net ecosystem $CO_2$ exchange (NEE). The gross primary production (GPP) was derived by subtracting the modelled respiration (R) from the NEE observation (GPP=NEE-R) utilizing standard flux partitioning procedures (Reichstein et al., 2005; Kolari et al., 2009). By using the same parameterisations as in the partitioning, the NEE and GPP time series were gap-filled for comparison with the model results. The daily evapotranspiration (ET) sums were calculated from $H_2O$ flux data that were gap-filled based on the mean diurnal cycles or regressions on available radiative energy

The EC instrumentation in Hyytiälä consisted of a Solent 1012R3 three-axis sonic anemometer (Gill Instruments Ltd., Lymington, UK) and a LI-6262 closed-path $CO_2$/$H_2O$ gas analyser (Li-Cor Inc., Lincoln, NE, USA), while in Sodankylä a USA-1 (METEK GmbH, Elmshorn, Germany) anemometer and an LI-7000 (Li-Cor., Inc., Lincoln, NE, USA) closed-path gas analyser was used. The EC fluxes were calculated as half-hourly averages taking into account the required corrections. The measurement systems and the post-processing procedures have been presented in more detail for Hyytiälä by Kolari et al. (2004) and Mammarella et al. (2009), and for Sodankylä by Aurela (2005) and Aurela et al. (2009).

The measurement error in the EC flux data may be classified into two categories: systematic errors and random errors. The main systematic errors (density fluctuations, high-frequency losses, calibration issues) are mostly corrected for as part of the post-processing of the data, and the random errors tend to dominate the uncertainty of the instantaneous fluxes. The random

error is often assumed Gaussian but can be more accurately approximated by a symmetric exponential distribution (Richardson et al., 2006). It increases linearly with the magnitude of the flux, with a standard deviation typically less than 20% of the flux Richardson et al. (2008); Rannik et al. (2016).

## 2.2 The JSBACH model

JSBACH is a process based ecosystem model and the land surface component of the Earth System model of Max Planck Institute for Meteorology (MPI-ESM). We used JSBACH offline using an observational atmospheric data set to force the model. Implications of this one-way coupling with the atmosphere include lack of feedback from the surface energy balance to the atmosphere, i.e. latent and sensible heat fluxes and surface thermal radiation do not directly affect prescribed air temperature or humidity. Similarly the feedback of surface to the vertical transfer coefficients within the atmospheric surface layer is missing as the wind speed that drives mixing is prescribed. Furthermore, since we use site level data (a single grid point), the grid resolution does not affect the results (Tesfa et al., 2014; Singh et al., 2015). We give here a general introduction to JSBACH whereas a more complete model description can be found in Roeckner et al. (2003).

In JSBACH the land surface is a fractional structure where the land grid-cells are divided into tiles representing the most prevalent vegetation classes called plant functional types (PFTs) within each grid cell (Reick et al., 2013). The grid cell is first divided into bare soil and vegetative area which is furthermore fractionally divided into PFTs. The model was setup to effectively use only one tile, coniferous evergreen trees. Henceforth all model and process descriptions are considered in relation to coniferous evergreen trees and no distinction between PFT specific and general parameters are made in this study.

Coniferous evergreen trees are characterized by a set of parameters that control vegetation related biological and physical processes accounting for the land-atmosphere interactions. We made use of expert knowledge to set these parameters for our local sites and verified that they are in line with those presented by Hagemann (2002); Hagemann and Stacke (2015).

The seasonal development of LAI is regulated by air temperature and soil moisture with a specific maximum LAI as a limiting value. The cycle is driven by a pseudo soil temperature that is a weighted running mean of air temperature. The predictions of phenology are produced by the Logistic Growth Phenology (LoGro-P) model of JSBACH.

Photosynthesis is described by the biochemical photosynthesis model (Farquhar et al., 1980). Following Kattge et al. (2009) we set the maximum carboxylation rate at 25 degrees Celsius to 1.9 times the maximum electron transport rate at 25 degrees Celsius.

The photosynthetic rate is resolved in two steps. First the stomatal conductance under conditions with no water stress is assumed to be controlled by photosynthetic activity (Schulze et al., 1994). Here the leaf internal $CO_2$ concentration is assumed to be a constant fraction of ambient concentration which allows for an explicit resolution of the photosynthesis (Knorr, 1997). Then the impact of soil water availability is accounted for by a soil moisture dependent multiplier that is identical for each canopy layer (Knorr, 1997).

Radiation absorption is estimated by a two stream approximation within a three-layer canopy (Sellers, 1985). Especially in the sparse canopies the radiation absorption is affected by clumping of the leaves which is here taken into account according to the formulation by Knorr (1997).

## 2.3 The JSBACH model spin up and runs

Before tuning the JSBACH model, some of the more slowly changing variables (e.g. LAI) need to be equilibrated in order to bring the model into a (semi)steady state. We achieve this by running the model through a spin up period generated by looping the measurement interval over itself. During this period the necessary variables are equilibrated and their values become acceptable for the tuning process. At the end of the spin up a restart file is generated so that the model can be restarted from this state.

We use half-hourly measurements from years 1999–2008 for Hyytiälä. The spin up finishes at the end of 1999 and is followed by a calibration period (abbreviated as HC for Hyytiälä calibration) of 2000–2004 and a validation period (HV) of 2005–2008, including an exceptionally dry summer in 2006. The setup for Sodankylä is similar but we use measurements from 2000–2008, where the spin up finishes at the end of 2008. The model is then restarted from the start of 2000 but we only examine the Sodankylä validation period (SV) of 2005–2008. The main reason to exclude the Sodankylä calibration period is that essentially we do not calibrate (or tune) the model for Sodankylä and we do not want to appear to do so.

The meteorological data used to drive the climate were air temperature, air pressure, atmospheric $CO_2$ concentration, precipitation, specific humidity, short- and longwave radiation, potential shortwave radiation and wind speed.

## 2.4 The parameters

The JSBACH model was modified to fit our custom-built testbed so that all parameters of interest could be read from an external file. We examined 15 parameters (Table 1) that are for convenience separated into three classes. The class I parameters are used differently from those of class II and III – namely class I parameters are only tuned in the seasonal tuning (explained in detail in chapter 3.1). Additionally the only destinction between class II and III parameters is that the latter belong to a specific part of JSBACH called the Logistic Growth Phenology model (LoGro-P) – there is no difference in how these parameters are used. We also note that the only parameter (of those examined) that can vary from site to site is $veg_{max}$ (the vegetative fraction of a grid cell).

## 2.5 Parameter sampling

The parameter sampling in this study was done with the Adaptive Metropolis (AM) algorithm. The AM algorithm is an adaptive Markov Chain Monte Carlo (MCMC) process described below (it is not strictly Markovian but satisfies the necessary ergodicity requirements). AM is based on the classical Metropolis algorithm, extended with the adaptation of the parameter proposal distribution. Due to the adaptive nature of AM, it does not rely on the choice of the initial proposal distribution. AM is a sampling method that produces estimates of the full distribution of possible parameter values (unlike straightforward optimiztion methods), thus enabling e.g. the study of parameter identifiability, sensitivity and (nonlinear) correlation – this information is paramount to understanding the optimization process in contrast to merely receiving the optimized parameter values. The rigorous mathematical presentation of the AM algorithm is given in Haario et al. (2001).

The AM algorithm draws samples (sets of parameters) from the parameter space to generate probability distributions for the parameters. The consecutive draws form an MCMC chain. We used the algorithm simultaneously for several independent chains that are parallel adaptations of the algorithmic process (see e.g. Craiu et al., 2009; Solonen et al., 2012) – we take several random starting points and launch the algorithm for each of these simultaneously. The history of all chains is used for updating the proposal covariance matrix that describes how the parameters relate to one another. Our initial proposal covariance matrix had diagonal elements corresponding to 1/200 of the respective parameter's range. The first sample for each chain was chosen at random within this range. The algorithmic process can be described with few steps:

1. Draw a new sample (*x'*) of the parameter space from the vicinity of the current sample (*x*) using the current proposal covariance matrix.

2. Calculate the acceptance ratio (*a*) for the drawn sample. This is the value of a likelihood function (*f*), that is proportional to the desired probability distribution, at the drawn sample divided by the value at the current sample ($a = f(x')/f(x)$).

3. Accept the new candidate (*x'*) with the probability *a* (if $a \geq 1$, we always accept). If the candidate was rejected, the current sample (*x*) is reused as a basis of the next draw and repeated in the chain. Update the covariance matrix and draw a new sample.

We obtain the likelihood function (*f*) from the cost functions (*cf*) described below by assuming gaussian error statistics and setting $f = e^{-cf}$. In general to estimate the distribution of parameters of any model based on some data, we require some information about the underlying measurement and modelling errors. We treat the JSBACH model as described by the equation $\mathbf{y} = M(\mathbf{x}, \theta) + \mathbf{e}$. Here $\mathbf{y}$ are the observations, $\mathbf{x}$ is the model state vector, $\theta$ are the current parameters and $\mathbf{e}$ is the model-data mismatch. Since we only have a robust estimate for the measurement errors and no true error statistics for the model, the full error ($\mathbf{e}$) is treated as gaussian white noise.

The cost function (1) used in this study for seasonal tuning is based on summary statistics of gross primary production (GPP) and evapotranspiration (ET) along with the maximum of leaf area index (LAI). Cost function (1) calculates the relative error in total GPP, ET and growing season maximum of LAI against observations (these are respectively denoted as $G_1$, $E_1$ and $L_1$) and sums them up. Overlined variables refer to the mean value of that variable for a given period (calibration or validation), subscripts denote observation or model result.

$$cf_1 = \overbrace{\left(\frac{\overline{GPP}_{mod} - \overline{GPP}_{obs}}{\overline{GPP}_{obs}}\right)^2}^{G_1} + \overbrace{\left(\frac{\overline{ET}_{mod} - \overline{ET}_{obs}}{\overline{ET}_{obs}}\right)^2}^{E_1} + \overbrace{\left(\frac{\max(LAI_{mod}) - \max(LAI_{obs})}{\max(LAI_{obs})}\right)^2}^{L_1} \tag{1}$$

The second cost function (2) is a slightly modified mean squared error estimate used for daily ($cf_2$) and half-hourly ($cf_3$) tuning. With multiple variables there is always the problem of having one variable dominating over the others. Since no true errors were available, it was decided to normalize the residuals using the mean of observations in cost function (2). This way the resulting function is sensitive to changes in both variables – AM is used as a noise-resistant optimizer and sampling is done

in the spirit of studying the identifiability and correlations of the parameters. The components are denoted as $G_2$, $E_2$ for daily and $G_3$, $E_3$ for half-hourly tuning.

$$cf_{2,3} = \overbrace{\frac{1}{N_{GPP}} \sum \left( \frac{GPP_{mod} - GPP_{obs}}{\overline{GPP}_{obs}} \right)^2}^{G_{2,3}} + \overbrace{\frac{1}{N_{ET}} \sum \left( \frac{ET_{mod} - ET_{obs}}{\overline{ET}_{obs}} \right)^2}^{E_{2,3}} \tag{2}$$

As noted previously JSBACH was used uncoupled from the other components of the full MPI-ESM. This has a tendency to lead to biased results in the model runs as has been recently studied by Dalmonech et al. (2015). Especially in the high latitudes evapotranspiration can be unrealistic during winter since night-time is longer and temperatures low. In order to improve the

5 credibility of our results, we masked the evapotranspiration values of the coldest periods, and only took into account those from May to September for each year in both cost functions.

## 2.6 Parameter analysis

The optimized parameter values are taken as the mean values of all chains in the sampling process. In a case that the parameter chains converge to a bound of an a priori prescribed range of allowed values, the maximum a posteriori (MAP) value is used

instead. After tuning the model, we analysed different aspects of this process. Class I parameters are excluded from this analysis since they are used to bring the model to an "acceptable initial state" hence we regard them as a part of the model initialization (our motivation is explained in chapter 3.1).

We calculated the correlations and correlation matrices between different parameters for different tunings using the tested parameter vectors in the AM process. Then we performed a principal component analysis (PCA) on the correlation matrices

to get the eigenvectors ($v_i$) and eigenvalues ($e_i$) of the least identifiable parameters in the tuning process with the given data. The PCA transforms the correlation matrix into an orthogonal form where the eigenvector related to the greatest eigenvalue is the least identifiable with the given data. We then calculate the weight ($w_i = \sqrt{\frac{e_i^2}{\sum_i e_i^2}}$) for each component (or vector $v_i$, note that the squared weights sum up to one). We also determine the most dominant parameters for each component ($v_i$) by similar dividing the length of the vector towards that parameter by the length of the whole vector (weight of vector components).

The information derived with PCA could be extracted by analysing the parameters posterior probability distributions but PCA yields a simple, straightforward method for the same purpose. The main caveat of the standard PCA method is that it is not applicable to cases with strong nonlinear correlations. Therefore we also calculate kernel density estimates (KDE) for the parameters to show that there are no nonlinear correlations. The KDE method places a gaussian distribution (kernels) centered at each parameter of the MCMC chain and then sums these kernels to produce an estimate for the whole distribution. The

bandwith is calculated using the Scott's rule (Scott, 2004).

We also wanted to examine which parameters contributed the most to the change in the cost function values as we switched from one parameter set to another. This was done by calculating the change in the cost function values of the tuned parameter set and a set where one parameter has been reverted to the value the tuning started with (hencefort the reference value – for seasonal tuning the default values and for daily and half-hourly tunings the seasonally tuned values). We call this method

here "relative effectiveness" since we want to analyse the effect of the parameters to the cost function. For each tuned set of parameter values, the relative effectiveness of a parameter is calculated as follows:

1. Change one parameter from the set of tuned parameter values to a reference value and calculate the difference in the cost function for the changed set and the tuned set.

2. Return the changed parameter to the tuned value and repeat for all parameters. Sum up the differences.

3. The relative effectiveness for each parameter is the difference obtained from step 1 divided by the sum from step 2.

The relative effectiveness is similar to a class of methods commonly referred to as one-at-a-time (OAT) or one-factor-at-a-time (OFAT). These methods are generally used to acquire robust information about model behaviour when one parameter

at a time is changed to a new and hopefully a better value (e.g. Murphy et al., 2004). The main difference of our method to classical methods such as Morris OAT (Morris, 1991) is that in such methods the change in values is (usually) random, where as we have fixed values. Additionally our point of view is from the optimized parameters to the original state – we have already optimized the parameters (as a group) and merely want some robust and easily comprehensible information about the effect of changes in parameter values to the cost functions. This method does not reveal information about how well the parameters

constrain the cost function (e.g. we could have a highly dominating parameter that would optimize to the default value and hence the relative effectiveness would be zero), rather which parameters contribute most to the change in cost function values.

Lastly we calculate the root mean squared error (RMSE, $\sqrt{\sum_i \frac{(o_i - m_i)^2}{n}}$), bias ($\sum_i \frac{o_i - m_i}{n}$) and the coefficient of determination ($r^2 = 1 - \frac{\sum_i (o_i - m_i)^2}{\sum_i (o_i - \overline{o_i})^2}$) for the time series generated by the different tunings ($o_i$ is observed and $m_i$ is modelled).

## 3 Model tuning

The model was optimized for Hyytiälä with the AM algorithm using three different time scales: seasonal, daily and half-hourly tuning, which are described below. Tuning was done on a powerful laptop with an Intel Core i7-3520M processor. We removed unwanted output streams from the model and tweaked the I/O. With a single core the spin up generation takes approximately 150 seconds, the run through calibration period with daily output takes 20 seconds and with half-hourly output 320 seconds. We used daily output also for the seasonal tuning.

### 3.1 Seasonal tuning

The fundamental motivation for the seasonal tuning is to ensure that the model reproduces the observed growing season maximum of LAI since we have previously noticed that JSBACH underestimates LAI at the site level (even the default value of $\Delta_{max}$ is lower than the measured maximum for Hyytiälä). The reason for this approach was to enhance the vegetation transpiration and to emphasize the model response to precipitation. We also want to ensure that the model performs adequately

well in terms of seasonal cumulative GPP and ET. The seasonal tuning was done in three consecutive steps each using the cost function (1). The procedure is as follows:

1. Tuning of all three class I parameters with four independent chains each consisting of 3000 samples. This step required a 30-year spin up for each sample separately.

2. Testing of class II and III parameters each separately with 24 evenly separated values for an extensive range and tuning those nine parameters that didn't yield a negligible difference in the maximal and minimal values in the objective function. The consequent tuning was done with eight independent chains each consisting of 10 000 samples. A single spin up, common for all samples, used optimal parameter values from step 1 and default values for the rest of the parameters.

3. Retuning all the previously tuned 12 parameters with eight independent chains each consisting of 10 000 samples. Initial proposal covariance was generated from previous step and spin up was generated separately for each sample.

At the end of seasonal tuning, class I parameters were fixed and a single spin up was generated to be used with daily and half-hourly tuning. This approach is computationally justifiable (as we do not have to rerun the spin up at each iteration of the algorithm) and is also acceptable from a modelling point of view since the robust site level scaling has already been done. Vegetative fraction of a grid cell remained at its default value of 0.52 and carboxylation rate at 25 degrees Celsius was lowered to 45.0 (and electron transport rate to 85.5).

## 3.2 Daily and half-hourly tuning

The difference in daily and half-hourly tuning is the time interval used in the model output and observations in the cost function (2). For both tuning runs we first tested the response of class II and III parameters against the cost function (2) and removed those parameters that yielded only negligible or no response (as in step 2 in Seasonal tuning). The rest of the parameters (twelve) were then tuned using eight independent chains each consisting of 10 000 samples.

It should be noted that even though the cost function (2) formulation is the same for daily and half-hourly tuning, the values of the cost function are not directly comparable. Half-hourly tuning uses 48 values per day, and the resulting diurnal pattern resembles the form of the normal distribution. In daily tuning we use an average of these values. In practice the component and cost function values will be higher for half-hourly tuning.

### 3.3 Tuning for Sodankylä

After tuning the model for Hyytiälä we took the parameter set from seasonal tuning and retuned only the maximum LAI parameter ($\Delta_{max}$) with the cost function (1) for Sodankylä. This was done because the measured LAI for Sodankylä is approximately half of that of Hyytiälä. The other parameter values were taken from the respective Hyytiälä tuning runs and spin ups were generated similarly to Hyytiälä spin ups so that we could use the Sodankylä results to validate the tuning process.

## 4 Results and discussion

The parameters and cost function components involved in the different phases of the optimization process need to be studied before the performance of the optimization method can be evaluated.

As noted above, we decided to reject the unreliable wintertime ET values. This masking leaves out the start of the growing season, which reduces the reliability of some of the tuned parameters, including all the LoGro phenology model parameters (class III), which mostly affect the timing of the spring event and regulate the development of the LAI towards the peak season. However, as a result of the tuning processes, all the analysed parameters were revealed to have unimodal posterior probability distributions, with different skewness's and deviations.

We analysed the correlations and effectiveness of the parameters in the seasonal, daily and half-hourly optimizations on the Hyytiälä site for the calibration period. We also analysed the contributions from the cost function components referring to ET, GPP and LAI and generated the time series and daily cycles of GPP and ET for both Hyytiälä and Sodankylä sites. For all these examinations, individual spin ups were generated using the optimized parameter values.

The parameter correlations (Table 2) do not reveal much information, which is common for larger systems where the underlying parameter dependencies are more complex. Usually more sophisticated methods are used to analyse the parameters, but we omit these examinations here since pairwise Kernel density estimates (Fig. 1) did not reveal any new insights.

The strongest correlation was between the ratio of leaf internal $CO_2$ concentration to external $CO_2$ ($f_{C3}$) and fraction of soil moisture above which transpiration is unaffected by soil moisture stress ($w_{tsp}$) in all the tunings. This positive correlation strengthens as we increase the temporal resolution (and the complexity of the underlying cost function). This is due to the carbon assimilation being limited by the amount of carbon available but also by a linear water stress factor (which takes the value of zero at the wilting point ($w_{wilt}$) and one at the $w_{tsp}$), which is checked at each time step. Most of the other parameters with high correlations are those of the LoGro phenology model, where we would expect high correlation since the parameters are intimately connected.

Approximately half of the parameters with high correlation are also the least identifiable (Table 3) with the given data and cost function. This means that the values these parameters acquire, as a result of the tuning process, are the most unreliable – it does not reflect on the parameters contribution to the cost function. The PCA merely highlights where most of the parametric unreliability lies.

The PCA analysis revealed that most of the unreliability is explained by a handful of parameters. Disregarding those of the LoGro phenology model, the two most dominantly unreliable parameters in every tuning were the fraction depicting relative humidity based on soil dryness ($w_{hum}$) and the maximum field capacity of the skin reservoir ($w_{skin}$). Both of these parameters affect the amount of water available for evaporation from bare soil and are both subject to changes in other parameters. Bare soil evaporation is also dominated by vegetative transpiration, which explains why these two parameters are the most unreliable.

## 4.1 The parameters and their relative effectiveness

The default and optimized parameter values from the different tuning metrics are presented in Table 4 along with their relative effectiveness. The reference values for seasonal tuning are the default values. Since we fixed class I parameters with seasonal tuning, the realistic reference values for daily and half-hourly tunings are the seasonal parameter values. Here we note that using one spin up for all daily and half-hourly optimization runs is computationally justifiable but generates errors as the

general spin up differs from those generated by the optimized parameters. These errors are relatively small but give rise to e.g. the negative relative effectiveness values in daily and half-hourly parametrizations.

Most seasonally tuned parameters are near their default values and the most effective parameters are the fraction of soil moisture above which transpiration is unaffected by soil moisture stress ($w_{tsp}$), the fraction of soil moisture at permanent wilting point ($w_{pwp}$) and the fraction of field capacity above which fast drainage occurs ($w_{dr}$). For daily and half-hourly tunings the most important parameters are the ratio of leaf internal $CO_2$ concentration to external $CO_2$ ($f_{C3}$) and the fraction depicting relative humidity ($w_{hum}$). It should be noted that $w_{hum}$ was one of the least identifiable parameters for seasonal tuning. Taking into account the importance of these parameters on transpiration and soil moisture estimations, we took a closer look at modelled soil moisture and evapotranspiration components for the calibration period (taking into account only values from May to September for each year as explained in chapter Uncoupled model runs).

When we compare the model output streams with seasonal against those with default parametrization, we notice that the average evapotranspiration for the calibration period has increased 15%. Most of this is due to added transpiration (18% increase) but also increased evaporation (6%). In addition drainage was accelerated by 11%. These increases are mostly compensated by a 15% reduction in average soil moisture. In addition soil moisture values that are under the limit when transpiration is affected by soil moisture stress (below the value of $w_{tsp}$) increased 2.3%.

The daily and half-hourly tunings lower the average evapotranspiration by 22% and 35% respectively when compared to the seasonal values. Transpiration is decreased by 28% and 37% whereas evaporation is increased by 0.5% and decreased by 28%, respectively for daily tuning and half-hourly tuning. Soil moisture is increased by 11% and 8% and the amount of values below $w_{tsp}$ is decreased by 62% for daily tuning and increased by 7% for half-hourly tuning. As a curiosity, both the adjustment parameter in stability functions ($c_b$) and the fraction of precipitation intercepted by canopy ($p_{int}$) have been significantly increased with daily tuning and returned to seasonally tuned values with half-hourly tuning.

## 4.2 The cost function components

Using the optimized values (parametrizations) we calculated the components of each cost function for Hyytiälä calibration period and Hyytiälä and Sodankylä validation period (Table 5).

Firstly we note that with the default parameters $L_1$ dominates $cf_1$ for Hyytiälä and contributes to approximately 90% to its value. As expected the $L_1$ for Sodankylä is not as dominant as for Hyytiälä since the measured maximum of LAI for Hyytiälä is roughly half as large as for Sodankylä, which directly lowers the LAI component in cost function (1). The $L_1$ contribution is significantly reduced with the seasonally tuned parameters as was our intention and even though LAI plays no part in daily and half-hourly tunings, the differences in the maximum value are negligible.

Secondly the value of $E_1$ component (error in seasonal ET) with default parametrization is significantly increased in daily and especially half-hourly parametrizations. Simultaneously the value of $G_1$ is significantly lowered. The component values for seasonal parametrization are better than the default values with the exception of $E_1$ for Hyytiälä validation period.

Thirdly for cost function (2) the pairwise ratio of dominating $E_i$ or $G_i$ components in all tunings is 5:1. On average $E_2/E_3$ contributes to approximately 60% of $cf_2/cf_3$. This translates to ET being twice as significant as GPP in cost function (2). The

main reason for ET dominating GPP is that ET is more erratic in comparison to GPP and the residuals of ET (divided by the mean value) are larger than the residuals of GPP. The daily and half-hourly tunings themselves work as intended as they lower the corresponding cost function value. It is noteworthy to mention that the $G_2$ component gets its lowest value for both validation periods with the half-hourly parametrization even though $G_2$ calculates GPP errors on a daily scale.

Lastly we examine how the algorithm and cost functions have performed. The best parameter set (lowest cost function value) for a given cost function, in each of the three different periods (HC, HV, SV), is the same that was used in the corresponding tuning process. For example the lowest value for $cf_1$ (cost function for seasonal tuning) in Sodankylä validation period (0.07) coincides with the seasonally tuned parameters. This is expected as the tuning process aims at the "best" parameter value, which reassures us that no gross mistakes (human errors) have been made. The relation holds true for every cost function with the exception of $cf_1$ for Hyytiälä validation period, where the lowest value is reached with the daily tuned parameters (we note that the absolute difference between daily and seasonally tuned parameters is small). Hence we can confidently state that the algorithm and cost functions have performed as intended, especially since the optimised parameters work for Sodankylä as well, where no optimization (besides the site specific maximum of LAI) was applied.

## 4.3 Time series

The overall structure of the model time series was not affected by the parametrizations obtained with different tunings (Fig. 2 and Fig. 3). Some time series characteristics have been enhanced and others reduced but the timing of the peaks and dips in GPP and ET are the same as before. The corresponding RMSE and bias estimates are given in Table 6. In comparison we estimated the PRELES model biases for Hyytiälä from Fig. 5 in Peltoniemi et al. (2015b). These estimates give a bias of 0.81E-6 kg m$^{-2}$s$^{-1}$ (0.07 mm m$^{-2}$d$^{-1}$) for ET and -1.45E-7 mol[CO$_2$] m$^{-2}$s$^{-1}$ (-0.15 g(C) m$^{-2}$d$^{-1}$) for GPP. Additionally the coefficient of determination ($r^2$) for GPP in Hyytiälä is in range of 0.74–0.76 for all tunings whereas the values reported in literature range from 0.68 (Trusilova et al., 2004) to 0.96 (Peltoniemi et al., 2015b) with most above 0.9 (Aalto et al., 2004; Duursma et al., 2009). For additional comparisons see also e.g. Abramowitz et al. (2007). Note that our estimates are calculated using only values from the start of May to the end of September.

The best seasonal performance was obtained by seasonal tuning as we previously noticed from the cost function components (Table 5). Even though the optimization is done on the seasonal level, especially the GPP cycle is noticeably improved from that generated by the default parameters. This tuning also gives rise to the most stable (least fluctuating) water use efficiency (WUE), when calculated as a pointwise ratio of GPP and ET. We use WUE here only as a diagnostic variable to examine the balance between the GPP and ET.

When compared to the seasonal tuning, the daily tuning is worse on the seasonal scale and lowers both the ET and GPP cycles. WUE follows the observations better but starts to give rise to some fluctuation. With half-hourly tuning this behaviour is further enhanced and especially ET is lowered to too low levels which manifests the high WUE values. The worsening in the model time series with daily and half-hourly tunings are explained by biases in the diurnal cycle.

## 4.4 Diurnal cycles

Average diurnal cycles with different parametrizations (Fig. 4) show that modelled night-time ET values are too low when compared to the observed and this behaviour was not affected by the tunings. Low night-time values are compensated by too high midday values in the default and seasonal tuning so that the average daily and seasonal values are on an acceptable level. For the daily and half-hourly tuning, the algorithm lowers the daytime values, which results in too low average daily and half-hourly values. It is noteworthy to mention that with the default setting we get too low GPP for Hyytiälä but too high for Sodankylä. The unrealistic wintertime and the biased night-time ET values actually have the same origin. Since we do not have the coupling from the land surface model (LSM) back to the atmosphere, we get an erroneous energy balance as we lose the energy released by condensation.

Disregarding the default parametrization we notice that seasonal parametrization show the highest values, daily in the middle and half-hourly show the lowest values. Daily parametrization reproduces the observations for average diurnal cycle better than the others in every occasion except the GPP for Sodankylä, where half-hourly tuning is better (verified by pointwise RMSE from the average diurnal cycle). We also notice that Sodankylä daily patterns, and to some extent Hyytiälä as well, are slightly out of phase. Our current understanding is that this is (at least partly) due to a slightly misaligned sensor (which can cause significant errors on high latitudes), measuring radiation fluxes. Fortunately this affects mainly the cost function for half-hourly tuning since it is the only one operating on the densest half-hourly timescale.

## 4.5 Dry event

Dry period in the summer 2006 can be clearly located by the massive drawdown in observed GPP, and to a lesser extent in ET, at Hyytiälä (Fig. 2). In a closer look at this event (Fig. 5) it is evident that none of our parametrization schemes were able to capture it correctly. As it was with the time series, the overall structure of the daily time series during this event remains the same (there are no divergent aspects in the model output between the different tunings).

During the drought event (defined here as 31.7.–15.8.2006) the soil moisture is on average 27% lower for default, daily and half-hourly tuning and 40% lower for seasonal tuning when compared to the corresponding values from other years – seasonal tuning has the lowest overall soil moisture. During this event the modelled soil moisture decreases monotonically for all tunings and reaches the lowest values on 13th of August, after which it starts to rise. During the period the modelled ET and GPP are predominantly higher than the observations. WUE on the other hand follows the "observations" remarkably well and deviates from the observed only towards the end of the event when modelled ET drops to near zero values, coinciding with the lowest modelled soil moisture values. Gao et al. (2016) examines deviation in the dependencies of GPP and ET to vapour pressure deficit (VPD) between model and observation results under the most severe soil moisture stress conditions at the end of the prolonged period of soil water scarcity (that occurred in 2006). This can be attributed to the lack of explicit dependence of the modeled stomatal conductance on the atmospheric humidity.

## 5   Conclusions

Initially we tuned the model to produce near measured seasonal ET, GPP and especially maximum LAI to enhance the vegetation transpiration and to emphasize the response to precipitation. This was done successfully with seasonal tuning in the hopes of bringing forth the underlying model responses to dryness. With the consecutive daily and half-hourly tunings, we managed to improve the average diurnal cycles of both ET and GPP, but failed in reproducing the low ET and GPP levels during the dry event in 2006. Effectively we first (seasonal tuning) transferred water from soil moisture into (too high levels of) ET, and later (with daily and half-hourly tunings) transferred some of it back.

In addition to the maximum LAI ($\Delta_{max}$) and maximum carboxylation rate ($V_{C,max}$), the most effective parameters in the seasonal tuning were the fraction of soil moisture above which transpiration is not affected by soil moisture stress ($w_{tsp}$) and the critical fraction of field capacity above which fast drainage occurs for soil water content ($w_{dr}$). The reduction in ET and GPP was mostly accounted for by lowering the approximate ratio of leaf internal $CO_2$ concentration to external $CO_2$ ($f_{C3}$), which reduces the amount of carbon available for photosynthesis. For daily tuning ET was further reduced by the increase of the fraction of precipitation intercepted by canopy ($p_{int}$) and lower relative humidity fraction ($w_{hum}$ – air humidity is based on soil dryness).

Despite the fact that we were unable to enhance the dry response of the model, we are confident in saying that the algorithm itself worked well and performed as intended with the daily tuning providing the most reduction in model-data mismatch. We optimized twelve parameters simultaneously (with daily and half-hourly tunings) used eight fairly short chains (8000 samples). With daily tuning the resulting estimates are well matured, but with half-hourly tuning the parameter deviations are larger (which is probably due modelling inefficiencies and noise in measurements). Nevertheless all optimization procedures worked well in regards on what was optimized (seasonality, daily averages or diurnal cycle).

Recently Knauer et al. (2015) found canopy conductance formulation to be a key factor in prescribing the transfer of carbon and water between terrestrial biosphere and the lower atmosphere. Additionally Gao et al. (2016) found that during prolonged period of soil water scarcity, the lack of explicit dependence of the stomatal conductance on the atmospheric humidity is one of the contributing factors on this issue. Further studies into enhancing the dry response in JSBACH are needed and these studies should reflect these latest findings.

*Author contributions.*   T. Aalto, H. Järvinen, T. Markkanen and S. Hagemann chose the parameters in the optimization process and provided support througout the experiments. M. Aurela and I. Mammarella provided knowledge on the observations. J. Susiluoto provided the algorithm testbed and J. Mäkelä integrated the model into the testbed, ran the experiments and prepared the manuscript with contributions from all co-authors.

*Acknowledgements.*   This work was funded by the European Commission's 7th Framework Programme, under Grant Agreement number 282672, EMBRACE project, and the Nordic Centre of Excellence 'Tools for Investigating Climate Change at High Northern Latitudes' (eSTICC) under the Nordic Top-Level Research Initiative. This work was also supported by the Academy of Finland Center of Excellence (no.

272041), ICOS-Finland (no. 281255), and ICOS-ERIC (no. 281250) funded by Academy of Finland. This work used eddy covariance data acquired and shared by the FLUXNET community, including these networks: AmeriFlux, AfriFlux, AsiaFlux, CarboAfrica, CarboEuropeIP, CarboItaly, CarboMont, ChinaFlux, Fluxnet-Canada, GreenGrass, ICOS, KoFlux, LBA, NECC, OzFlux-TERN, TCOS-Siberia, and USCCC. The FLUXNET eddy covariance data processing and harmonization was carried out by the ICOS Ecosystem Thematic Center, AmeriFlux Management Project and Fluxdata project of FLUXNET, with the support of CDIAC, and the OzFlux, ChinaFlux and AsiaFlux offices.

## Appendix A:  Parametric equations within JSBACH

In this appendix we present the main equations that the parameters in this study affect.

### A1   Logistic Growth Phenology (LoGro-P) model

The parameters from the LoGro-P model that we are interested here, are mainly used to determine the spring event for JSBACH. The maximum all sided leaf area index ($\Delta_{max}$) is also part of this model, hence we introduce this first and then deal with the spring event. $\Delta_{max}$ is used to calculate LAI at each timestep by a logistic equation (A1). Here $k$ is the growth and $p$ the shedding rate, both of which further depend on temperature and soil moisture.

$$\frac{d\Delta}{dt} = k\Delta(1 - \frac{\Delta}{\Delta_{max}}) - p\Delta \tag{A1}$$

To determine the date of the spring event we first introduce a few additional variables, namely the heatsum ($S_T(d)$), the number of chill days ($C(d)$) and the critical heatsum ($S_{crit}(d)$). Also $T(d)$ denotes the mean temperature at day $d$.

$$S_T(d) = \sum_{d'=d_0}^{d} \max(T(d') - T_{alt}, 0) \tag{A2}$$

Heatsum $S_T(d)$ cumulates the amount of "heat" above the parameter $T_{alt}$ after the previous growing season. The actual starting date $d_0$ of the summation need not be known since it is enough to start the summation "reasonably late" after the last growth season.

$$C(d) = \sum_{d'=d_a}^{d} H\left(T_{alt} - T(d)\right) \tag{A3}$$

The number of chill days is calculated as the number of days when the mean temperature is below $T_{alt}$. Here $H()$ denotes the Heaviside step function and the summation starts at the day ($d_a$) of the last autumn event.

$$S_{crit}(d) = S_{min} + S_{range}e^{-C(d)/C_{decay}} \tag{A4}$$

The critical heatsum ($S_{crit}$) decreases as the number of chill days $C(d)$ increases. The spring event happens when:

$$S_T(d) \geq S_{crit}(d) \tag{A5}$$

Pseudo soil temperature ($T_s(t)$) at time $t$ is calculated as an average air temperature ($T$) with an exponential memory loss ($T_{ps}$). Pseudo soil temperature is used in determining the autumn event (when it falls below a certain treshold). In the equation $N$ is the normalization constant and $\tau$ is the length of a time step.

$$T_s(t) = \frac{1}{N} \sum_{n=-\infty}^{t} T(n)e^{-(t-n)\frac{\tau}{T_{ps}}} \tag{A6}$$

## A2 Photosynthesis

The Farquhar model is based on the observation that the assimilation rate in the chloropast is limited either by the carboxylation rate ($V_C$) or the transport rate ($J_E$) of two electrons freed during the photoreaction. The total rate of carbon fixation $A$ is given by the following equation, where $R_d$ is the so called dark respiration:

$$A = \min(V_C, J_E) - R_d \tag{A7}$$

Oxygenation of the Rubisco molecule reduces the carboxylation rate, which is given as:

$$V_C = V_{C,max} \frac{C_i - \Gamma_\star}{C_i + K_C(1 + O_i/K_O)} \tag{A8}$$

Here $C_i$ and $O_i$ are the leaf internal $CO_2$ and $O_2$ concentrations, $\Gamma_\star$ is the $CO_2$ compensation point, $K_C$ and $K_O$ are Michaelis-Menten constants parametrizing the dependence on $CO_2$ and $O_2$ concentrations. Furthermore leaf internal $CO_2$ concentration depends on the external concentration $C_E$ by:

$$C_i = f_{C3} C_E \tag{A9}$$

Likewise the electron transport rate is given as:

$$J_E = J(I) \frac{C_i - \Gamma_\star}{4(C_i + 2\Gamma_\star)} \tag{A10}$$

Here $J(I)$ is a function of radiation intensity $I$ in the photosynthetically active band, the maximum electron transport rate $J_{max}$ and the quantum efficiency for photon capture $\alpha_q$.

$$J(I) = J_{max} \frac{\alpha_q I}{\sqrt{J_{max}^2 + \alpha_q^2 I^2}} \tag{A11}$$

## A3 Soil water

In JSBACH the soil water budget is based on several reservoirs (e.g. skin, soil, bare soil, rain intercepted by canopy etc.) and the different formulations are plentiful. We present here only the most crucial of these. Changes in soil water ($w_s$) due to rainfall ($R$), evapotranspiration ($ET$), snow melt ($M$), surface runoff ($R_s$) and drainage ($D$) are calculated with a geographically varying maximum field capacity ($w_{fc}$).

$$\rho \frac{\partial w_s}{\partial t} = (1 - p_{int})R + ET + M - R_s - D \tag{A12}$$

The interception parameter ($p_{int}$) also affects the amount of water intercepted by vegetation and bare soil which further affects evaporation etc. The skin reservoir is limited by $w_{skin}$ and excess water is transferred to soil water. Likewise when the soil water content (in relation to maximum field capacity) is greater than parameter $w_{dr}$, the excess water is rapidly drained (in addition to the limited drainage below this threshold).

Evaporation from wet surfaces ($E_{ws}$) depends on air density ($\rho$), specific humidity ($q_a$), saturation specific humidity ($q_s$) at surface temperature ($T_s$) and pressure ($p_s$) and aerodynamic resistance ($r_a = C_h |v_h|^{-1}$, these are heat transfer coefficient and horizontal velocity).

$$E_{ws} = \rho \frac{q_a - q_s(T_s, p_s)}{r_a} \tag{A13}$$

Transpiration from vegetation ($T_v$) is likewise formulated but additionally depends on the stomatal resistance of canopy ($r$).

$$T_v = \rho \frac{q_a - q_s(T_s, p_s)}{r_a + r} \tag{A14}$$

The stomatal resistance is given as a minimal stomatal resistance of the canopy without water stress ($r_{min}$, depends on photosynthetically active radiation and LAI) divided by a water stress factor ($f_{ws}$). That is $r = r_{min}/f_{ws}$. The water stress factor depends on how much water is in the soil in relation to the maximum field capacity ($w_f = w_s/w_{fc}$) when compared to the limit when transpiration is no longer affected by soil moisture stress ($w_{tsp}$) and the permanent wilting point ($w_{pwp}$).

$$f_{ws} = \begin{cases} 1 & w_f \geq w_{tsp} \\ \frac{w_f - w_{pwp}}{w_{tsp} - w_{pwp}} & w_{pwp} \leq w_f \leq w_{tsp} \\ 0 & w_f \leq w_{pwp} \end{cases} \tag{A15}$$

Evaporation from dry bare soil ($E_s$) is similarly defined as:

$$E_s = \rho \frac{q_a - h q_s(T_s, p_s)}{r_a} \tag{A16}$$

Here $h$ is relative humidity at the surface relative to soil dryness:

$$h = \max\left[ w_{hum}(1 - \cos(\pi w_f)), \min\left(1, \frac{q_a}{q_s(T_s, p_s)}\right) \right] \tag{A17}$$

The total evapotranspiration is a weighted average of $E_{ws}$, $T_v$ and $E_s$, where the weights are based on e.g. fill levels of reservoirs (similar to $w_f$ above) and vegetative fraction of the grid cell ($veg_{max}$).

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

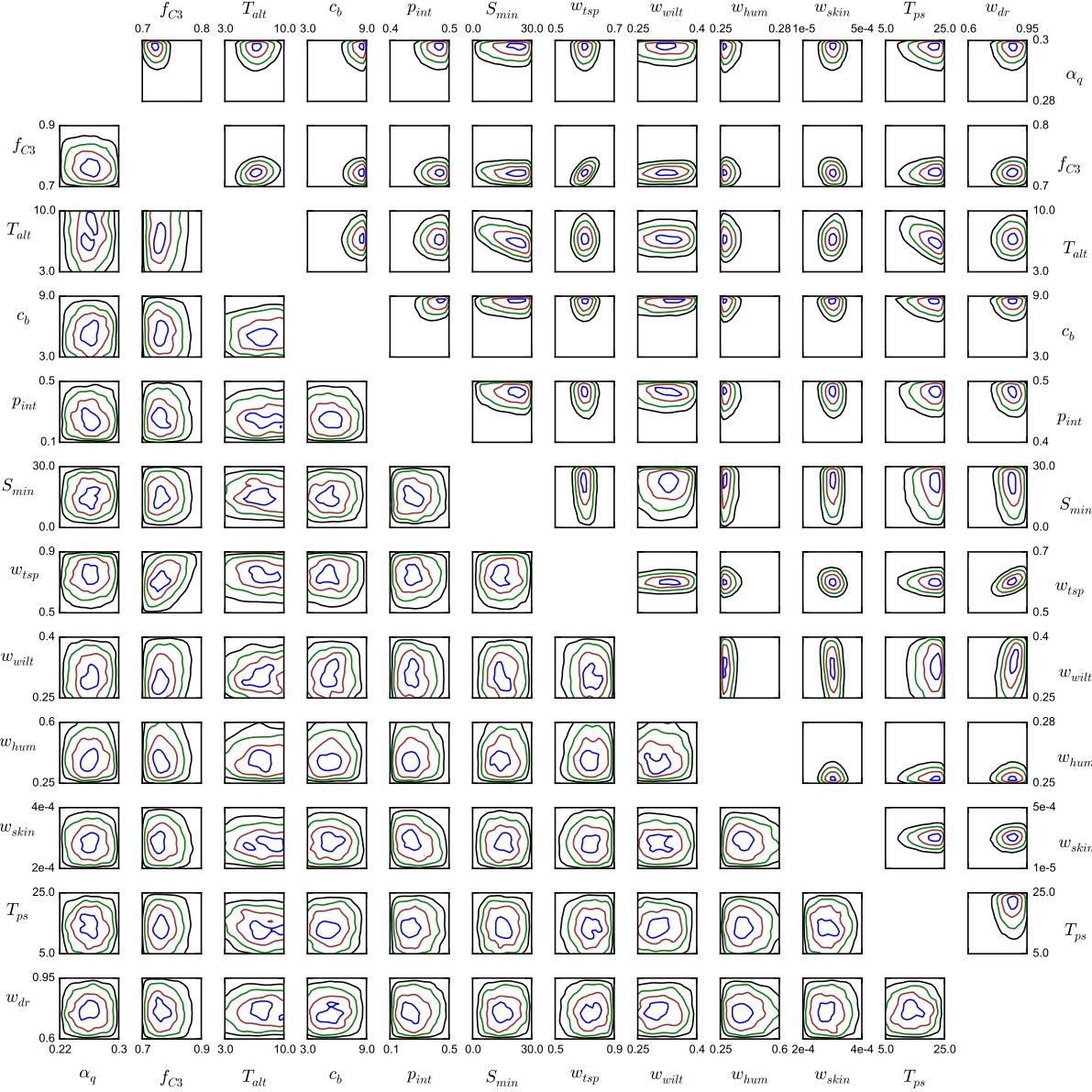

**Figure 1.** Kernel density estimates of the last 20 000 parameter samples with daily (upper triangle) and half-hourly tunings. The contours correspond to densities in a two dimensional gaussian distribution ($\mu_x, \mu_y = 0, \sigma_x, \sigma_y = 1$) with $2\sigma$ (black), $1.5\sigma$ (green), $\sigma$ (brown), $0.5\sigma$ (blue).

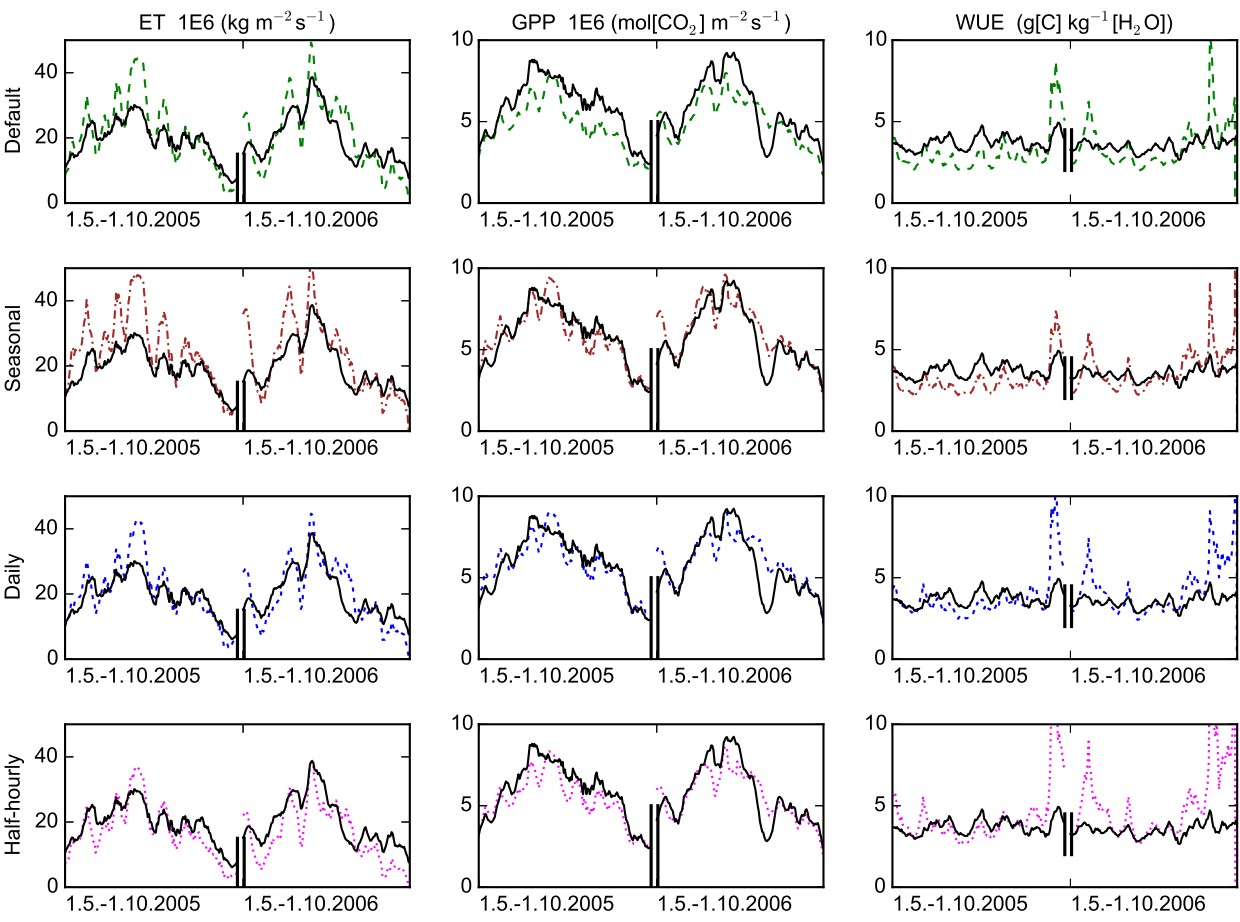

**Figure 2.** Hyytiälä 7-day running mean time series for different tunings for the first two summers of the validation period. Solid black line represents the observations.

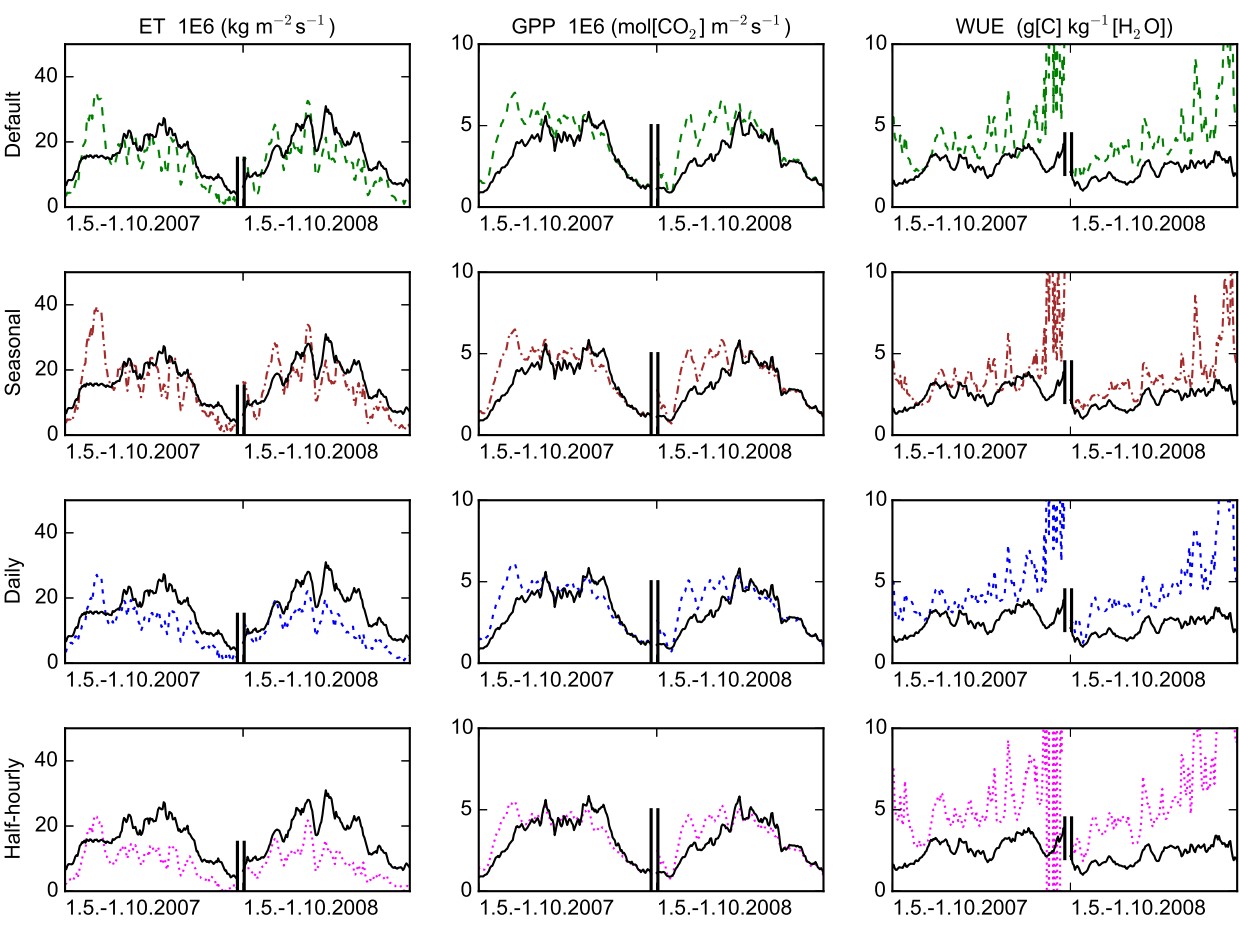

**Figure 3.** Sodankylä 7-day running mean time series for different tunings for the last two summers of the validation period. Solid black line represents the observations.

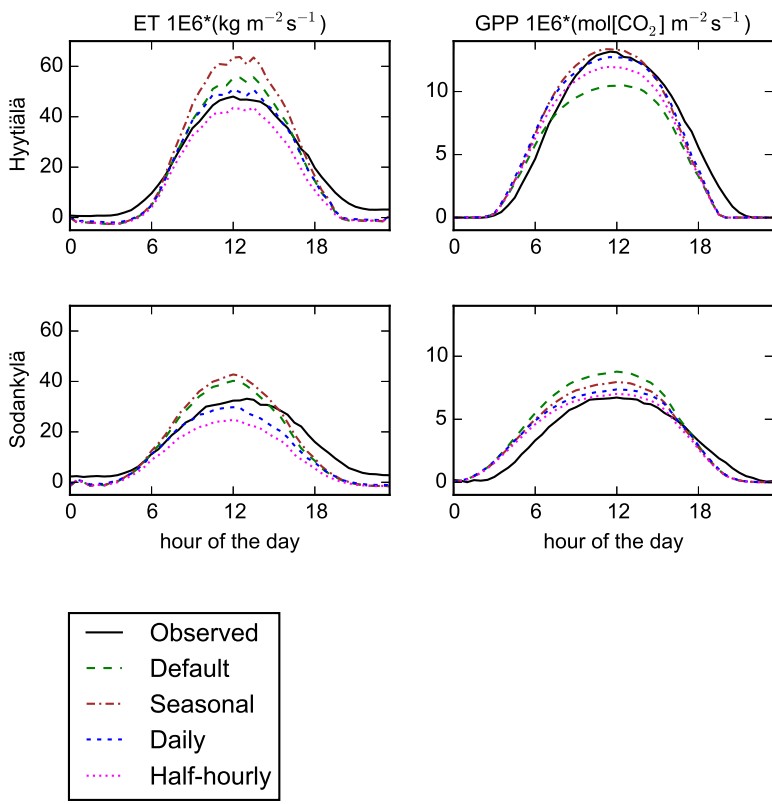

**Figure 4.** Average diurnal cycle from May to September for the validation period.

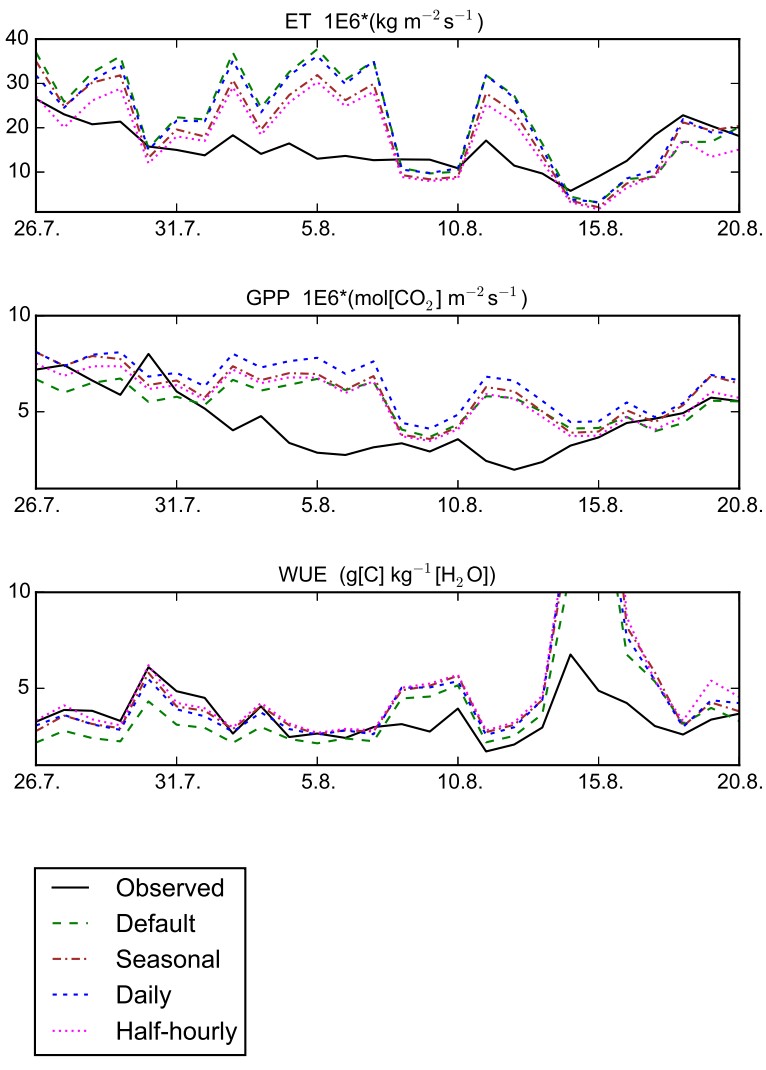

**Figure 5.** Daily averages for ET, GPP and WUE on a dry event in 2006 for Hyytiälä.

**Table 1.** Parameter descriptions with references to equations in appendix A. *These parameters were tested but yielded no or only minimal response to cost functions and were thus removed from the trial.

| Parameter | Units | Class | Desciption | |
|---|---|---|---|---|
| $\Delta_{max}$ | - | I | Maximum all-sided leaf area index that vegetation can reach. | A1 |
| $V_{C,max}$ | $\diamond$ | I | Farquhar model maximum carboxylation rate at 25°C of the enzyme Rubisco (coupled with maximum electron transport rate at 25°C with a factor of 1.9)    [$\diamond = \mu$ mol($CO_2$) m$^{-2}$ s$^{-1}$]. | A8 |
| $veg_{max}$ | - | I | Fraction of vegetative soil in a grid cell. The rest is bare soil. | - |
| $\alpha_q$ | - | II | Farquhar model efficiency for photon capture at 25°C. | A11 |
| $c_b$ | - | II | Adjustment parameter used in stability functions for momentum and heat (Louis, 1979). | - |
| $f_{C3}$ | - | II | Ratio of C3-plant internal/external $CO_2$ concentration. | A9 |
| $p_{int}$ | - | II | Fraction of precipitation intercepted by the canopy. | A12 |
| $w_{dr}$ | - | II | Critical fraction of field capacity above which fast drainage occurs for soil water content. | - |
| $w_{hum}$ | - | II | Fraction depicting relative humidity based on soil dryness. | A17 |
| $w_{pwp}$ | - | II | Fraction of soil moisture at permanent wilting point. | A15 |
| $w_{skin}$ | m | II | Maximum water content of the skin reservoir of bare soil. | - |
| $w_{tsp}$ | - | II | Fraction of soil moisture above which transpiration is not affected by soil moisture stress. | A15 |
| $s_{sm}$* | m | II | Depth for correction of surface temperature for snow melt. | - |
| $T_{alt}$ | °C | III | LoGro phenology: alternating temperature. Cutoff temperature used for calculating heatsum to determine the spring event and the number of chill days since the last autumn event. | A2, A3 |
| $C_{decay}$* | - | III | LoGro phenology: memory loss parameter for chill days. | A4 |
| $S_{min}$ | °C | III | LoGro phenology: minimum value of critical heat sum. | A4 |
| $S_{range}$* | °C | III | LoGro phenology: maximal range of critical heat sum. | A4 |
| $T_{ps}$ | °C | III | LoGro phenology: memory loss parameter for calculating pseudo soil temperature. | A6 |

**Table 2.** Highest correlations between parameters.

| Tuning | parameters | | r |
|---|---|---|---|
| seasonal | $f_{C3}$ | $w_{tsp}$ | 0.49 |
| | $T_{alt}$ | $\alpha_q$ | 0.40 |
| daily | $f_{C3}$ | $w_{tsp}$ | 0.52 |
| | $w_{dr}$ | $w_{tsp}$ | 0.52 |
| | $T_{alt}$ | $T_{ps}$ | -0.48 |
| | $T_{alt}$ | $S_{min}$ | 0.47 |
| half-hourly | $f_{C3}$ | $w_{tsp}$ | 0.68 |
| | $p_{int}$ | $w_{skin}$ | -0.44 |

**Table 3.** Significant components of principal component analysis for the different tunings. The given parameters are the most dominant within the component and ratio is how many times larger the factor related to the first parameter is when compared to that of the second. Coverage reveals how much of the component is accounted for by the given parameters (sum of the weights of given vector components).

| Component | weight | parameters | | ratio | coverage |
|---|---|---|---|---|---|
| seasonal 1. | 0.996 | $w_{hum}$ | $w_{skin}$ | 2.1 | > 99% |
| daily 1. | 0.717 | $T_{ps}$ | $w_{skin}$ | 1.4 | > 99% |
| daily 2. | 0.261 | $w_{hum}$ | $w_{tsp}$ | 2.3 | > 99% |
| half-hourly 1. | 0.530 | $T_{ps}$ | - | - | > 99% |
| half-hourly 2. | 0.310 | $w_{skin}$ | $w_{hum}$ | 1.7 | 96% |
| half-hourly 3. | 0.121 | $T_{alt}$ | - | - | > 99% |

**Table 4.** Default and optimized parameter values using the last 20 000 samples (if no value is given, the parameter was not part of that tuning and the default value was used instead). The percentage next to a parameter value is the effectiveness of that parameter for that tuning. The reference values for seasonal tuning are the default values and for daily and half-hourly tunings the seasonal values.

| Parameter | | default | seasonal | | daily | | half-hourly | |
|---|---|---|---|---|---|---|---|---|
| $\alpha_q$ | | 0.28 | 0.26 | 7% | 0.30 | 3% | 0.27 | 1% |
| $c_b$ | | 5.0 | - | - | 8.8 | 7% | 5.0 | 0% |
| $f_{C3}$ | | 0.87 | 0.88 | 8% | 0.72 | 70% | 0.76 | 68% |
| $p_{int}$ | | 0.25 | 0.27 | 1% | 0.49 | 4% | 0.27 | 0% |
| $w_{dr}$ | | 0.9 | 0.79 | 14% | 0.87 | 1% | 0.75 | -1% |
| $w_{hum}$ | | 0.5 | 0.54 | 1% | 0.25 | 14% | 0.37 | 22% |
| $w_{pwp}$ | | 0.35 | 0.28 | 10% | 0.34 | 0% | 0.31 | -1% |
| $w_{skin}$ | [m] | 2.0E-4 | 3.1E-4 | 6% | 3.0E-4 | 0% | 2.2E-4 | 6% |
| $w_{tsp}$ | | 0.75 | 0.64 | 53% | 0.60 | 1% | 0.75 | 3% |
| $T_{alt}$ | [°C] | 4.0 | 8.1 | 0% | 6.9 | 1% | 6.9 | 2% |
| $S_{min}$ | [°C] | 10.0 | - | - | 23.0 | -0% | 14.7 | -0% |
| $T_{ps}$ | [°C] | 10.0 | - | - | 21.0 | -0% | 12.4 | -0% |

**Table 5.** Cost function components for each parametrization for Hyytiälä calibration (HC), validation (HV) and Sodankylä validation (SV) periods. $L_1$, $E_1$ and $G_1$ are the LAI, ET and GPP components in cost function (1), represented by $cf_1$ and used for seasonal tuning. Likewise $E_2$ and $G_2$ are the components in cost function (2) for daily values ($cf_2$), whereas $E_3$ and $G_3$ are for half-hourly values ($cf_3$). Note that the values of $cf_2$ and $cf_3$ are not directly comparable.

|    |             | $L_1$  | $E_1$  | $G_1$  | $E_2$ | $G_2$ | $E_3$ | $G_3$ | $cf_1$ | $cf_2$ | $cf_3$ |
|----|-------------|--------|--------|--------|-------|-------|-------|-------|--------|--------|--------|
| HC | default     | 0.396  | 0.021  | 0.036  | 0.306 | 0.191 | 1.126 | 0.681 | 0.45   | 0.50   | 1.8    |
|    | seasonal    | 5.0E-5 | 1.7E-4 | 5.7E-6 | 0.343 | 0.161 | 1.326 | 0.720 | 2.3E-4 | 0.50   | 2.0    |
|    | daily       | 7.4E-5 | 0.055  | 1.4E-4 | 0.206 | 0.149 | 0.906 | 0.683 | 0.06   | 0.36   | 1.6    |
|    | half-hourly | 1.0E-4 | 0.128  | 5.4E-3 | 0.276 | 0.151 | 0.864 | 0.661 | 0.13   | 0.43   | 1.5    |
| HV | default     | 0.396  | 0.002  | 0.028  | 0.226 | 0.157 | 1.027 | 0.479 | 0.43   | 0.38   | 1.5    |
|    | seasonal    | 9.3E-5 | 0.011  | 7.5E-4 | 0.300 | 0.134 | 1.370 | 0.459 | 0.01   | 0.43   | 1.8    |
|    | daily       | 1.4E-4 | 0.007  | 3.5E-4 | 0.164 | 0.124 | 0.981 | 0.446 | 7E-3   | 0.29   | 1.4    |
|    | half-hourly | 1.1E-4 | 0.058  | 2.9E-3 | 0.182 | 0.118 | 0.748 | 0.412 | 0.06   | 0.30   | 1.2    |
| SV | default     | 0.108  | 4.0E-3 | 0.140  | 0.423 | 0.596 | 1.660 | 1.795 | 0.25   | 1.02   | 3.5    |
|    | seasonal    | 5.9E-3 | 1.8E-5 | 0.068  | 0.467 | 0.411 | 1.786 | 1.429 | 0.07   | 0.88   | 3.2    |
|    | daily       | 6.1E-3 | 0.063  | 0.048  | 0.289 | 0.352 | 1.258 | 1.294 | 0.12   | 0.64   | 2.6    |
|    | half-hourly | 5.9E-3 | 0.164  | 0.022  | 0.379 | 0.290 | 1.246 | 1.185 | 0.19   | 0.67   | 2.4    |

**Table 6.** RMSE and bias of ET and GPP calculated from half-hourly data for first two summers of validation period for Hyytiälä (corresponding to Fig. 2) and last two summers of validation period for Sodankylä (corresponding to Fig. 3).

|            | ET (kg m$^{-2}$s$^{-1}$) | | | | GPP (mol(CO$_2$) m$^{-2}$s$^{-1}$) | | | |
|------------|---------|----------|---------|---------|---------|----------|---------|----------|
|            | Hyytiälä | | Sodankylä | | Hyytiälä | | Sodankylä | |
|            | RMSE    | bias     | RMSE    | bias    | RMSE    | bias     | RMSE    | bias     |
| default    | 2.03E-5 | -1.31E-6 | 2.27E-5 | 2.31E-6 | 3.09E-6 | 8.77E-7  | 3.16E-6 | -9.19E-7 |
| seasonal   | 2.37E-5 | -4.32E-6 | 2.35E-5 | 1.09E-6 | 3.10E-6 | -2.00E-7 | 2.89E-6 | -5.97E-7 |
| daily      | 2.03E-5 | -0.74E-6 | 2.06E-5 | 5.00E-6 | 3.06E-6 | -1.07E-7 | 2.74E-6 | -4.57E-7 |
| halfhourly | 1.69E-5 | 2.77E-6  | 2.04E-5 | 7.14E-6 | 2.94E-6 | 3.39E-7  | 2.67E-6 | -2.79E-7 |