# Peer review of "Constraining ecosystem model with Adaptive Metropolis algorithm using boreal forest site eddy covariance measurements"

_Nonlinear Processes in Geophysics, 2016_

## Referee Comment (RC1) · Anonymous Referee #1 · 13 Jul 2016

General comments:

Parameters of the JSBACH land surface model are tuned for two forest sites in Finland. Photosynthesis and evapotranspiration estimates derived from eddy covariance measurements are used to calculate cost functions to be minimized. The optimization is able to correct for the main shortcomings of the model in the description of the annual cycle but is not sufficient to improve the representation of extreme events such as droughts. This shows that basic processes are missing in JSBACH. This kind of result is not new. The authors should do a better job in explaining what is new and original in their optimization approach. From a modelling perspective, a discussion is lacking about the reliability/robustness of JSBACH with respect to other models. From

a methodology point of view, several issues need to be clarified. Spin-up must be performed for any new set of parameter values and it is not clear whether the authors made this effort or not. The purpose of the parameter classification (class I, II, and III) is not clear. The classification itself is not properly described, nor justified. Although the paper is reasonably well written, part of the method description is found in the Result section and should be moved to the Methods section. The Abstract need to be improved.

Recommendation: major revisions.

Particular comments:

P. 1, Abstract: A summary of the main findings regarding the usefulness of the optimization technique used in this study is lacking. Key results and conclusions must be listed.

P. 3, L. 31: Why not including the spin-up into the calibration ? Please clarify.

P. 4, L. 7: Some Class II and Class III parameters can also be "site-specific". For example, soil water retension parameters are highly site-specific. Please clarify what you mean by "site specific".

P. 5, L. 11-12: This argument is not valid as some Class II and Class III parameters listed in Table 1 can be site-specific. Do you mean that Class I parameters are observed and do not need any analysis ?

P. 6, L. 22: Is using a single spin-up valid ?

P. 7, L. 8: Does this mean that class I parameters other than maximum LAI are not considered as site-specific ?

P. 9, L. 4: This paragraph is difficult to understand because the methods were not sufficiently described and symbols were not defined before. Methods, as well as "L1", and all the other symbols of Table 5 (including "HC", "HV", "SV") should be defined/presented in Sect. 2. Not here in the result Section.

P. 9, L. 6 ("half as large"): Half as large as what ?

P. 9, L. 14 ("ET is a more turbulent flux than GPP"): What do you mean ? GPP is not a turbulent flux at all. The turbulent CO2 flux is NEE, not GPP. GPP is not directly measured by eddy covariance techniques.

P.9, L. 27: The JSBACH model simulations don't look very good. How does JSBACH perform with respect to other models at these two sites ? Please give basic scores in terms of half-hourly fluxes, such as RMSD, ubRMSD and mean bias.

P. 11, L. 1: How can this be explained ? Shortcomings in the representation of the soil moisture stress ? How could these shortcomings be attenuated ? Using another photosynthesis model ?

Editorial comments:

P. 18 (Table 1): Parameters' units are lacking.

P. 19 (Table 4): Parameters' units are lacking.

P. 20 (Table 5, "highlighted values"): I don't see any highlighted value.

---

## Referee Comment (RC2) · Anonymous Referee #2 · 12 Aug 2016

Report on the article 'Makela, Susiluoto,Markkanen,Aurela,Mammarella,Hagemann, Aalto: Constraining ecosystem model with Adaptive Metropolis algorithm using boreal forest site eddy covariance measurements'

The work presents an approach for studying parameters of an ecosystem model. Especially, the focus is the site level parameter optimization of the JSBACH model, the land surface component of the Max Planck Institute MPI-ESM model. JSBACH simulates the water and carbon storages and fluxes of the ESM model. The motivation of the present work is to correct observed biases of the model, especially in evapotranspiration (ET) over continental areas and gross primary growth (GPP) under water limitation. The authors study how well the selected cost functions, featuring ET,GPP and LAI (maximum leaf area index) constrain the respective model parameters. The key idea is to use a Monte Carlo sampling method, adaptive MCMC, instead of a direct optimization of the parameters. This approach allows for a comprehensive study of the parameter identifiability. The authors are able to improve the model fit in several aspects, but can not remove the bias of an extremely dry season.

The work is certainly professional and worthwhile to publish. However, the novelty of the approach should be more clearly given, e.g., the benefits of the chosen MCMC approach should be clearly described. Also, a typical reader of the journal is most likely not too familiar with the details of the JSBACH model, and would need the authors giving more background and insight of the model and parametrizations of it. Some more specific comments below.

Major Comments

1) To estimate the distribution of parameters B of a model F based on data Y given by experiments X, connected by the standard expression 'Y= F(X,B) + eps', the distribution of the measurement error 'eps' should be known. But here the authors give almost no information of any of these to a reader not already familiar with JSBACH and the measurements. Certainly it is not possible to give all details, but the basic parts of the underlying modeling and numerical solution should be described, maybe in an Appendix, not to leave F(X,B) as a fully black box for the reader. See comments 4), 6) and 7) below.
2) To optimize the model parameters the Adaptive Metropolis (AM) method is chosen. It is, however, a sampling method rather than optimization. The motivation and benefits of the choice should be given: instead of a point estimate, samples of the full distribution of possible parameter values are obtained, together with (nonlinear) correlation information, sensitivity, identifiability of parameters, etc.
3) The parameter estimation is based on the two cost functions on p. 4 and 5. But no info is given here on the assumed statistics of the expressions, only a hint on Gaussian distribution later on p. 7. Usually, the sum of squares of the residuals is divided by the respective estimated variance of measurement error. Here, the residuals are normalized by the observations. This can be quite

acceptable if no 'true' error statistics is available, and the sampling is done in the spirit of studying the identifiability and correlations of the parameters. However, this should be done explicit in the text.

4) For the general audience (not familiar with JSBACH) at least the basics of the numerical approach used in JSBACH should be given, together with the CPU demands of the runs. Now only an implicit statement (' …interval is looped over to generate a 30 year spin up …', line 30 , p.3) is given that would indicate that JSBACH is a dynamic model that has to be initialized or run into a (quasi)steady-state to compare with observations ? Or is this due to the uncoupled version used here? The concept and use of spin-up should be clarified.

5) How much does the uncoupling impact the results in general? The authors mention (P.10, line 13-15) that the lack of coupling of the LSM model to atmosphere generates an erroneous energy balance. This aspect should be discussed or commented more explicitly.

6) The discussion in Section 2.5, parameter posterior distribution vs PCA, is not clear. The authors 'perform a PCA analysis transforms of the covariance matrices …' – but do not tell what covariance ? My guess would be that they actually mean the matrix of the AM samples of parameter vectors, and compute the PCA of it to get the eigenvectors of the least identified parameter directions. This can lead to correct conclusions, assuming that the nonlinear correlations between the parameters are not too strong. That, on the other hand, is typically indicated by plotting the 2D scatter plots of parameter marginal distributions. So I would recommend the authors to show them as well, and clarify the discussion on how PCA was used.

7) No information is given on how the studied parameters appear in the model. It is well-known that the parametrizations strongly impact the identifiability. A good example is the logistic function, where centering and scaling typically removes correlations. So this point should be made explicit by showing the formulas, at least in case of the LoGro phenology model where high correlations appear 'since the parameters are intimately connected' (L.30, p.7).

8) The measurements consist of the CO2 fluxes as given by the eddy covariance method. But the cost functions are given in terms of 'observed' and modeled GPP,ET and LAI. The connection between CO2 fluxes and those cost function expressions should be given.

Minor comments

1) In addition to the PCA/MCMC analysis of least identified parameters, the authors study which parameters are the most relevant for the change of the cost function. They introduce an OAT (one-at-a-time) method of their own (?). The relation of it to well-known methods such as the MOAT (Morris-OAT, see the reference below) could be make more clear. Also, it is not clear what the 'tuned parameter' (p.5, Step 1) is: the mean of the sampled values, or the maximum likelihood (minimum cost function) value? I would gather that the 'reference value' is the initial/default value of optimization. These points should be made clear.

2) Only the cost functions are given in the text, not the likelihood used in the sampling. If it is Gaussian as indicated on p.7, it should be mentioned that the 'f' function of step 2., p. 4, actually is the exponential function of the (negative) cost function.
3) P.4 line 15: The sentence 'A sample in the parameter has a value …' could be removed. Instead, the term 'chain' could be explained for a reader unfamiliar with MCMC.
4) P.4, line 16: edit the sentence 'The algorithm is used …' something like 'The algorithm can be used' or 'is used here', since the basic form of the AM algorithm is or a single chain. Maybe add a reference to parallel chain adaptive MCMC
5) P. 6, line 20/Step 3: 'Initial covariance' means the initial proposal covariance for MCMC sampling ?
6) P. 6, lines 17 and 27: it would be good to know here how many parameters were used for the 10000 sample long chains.
7) P. 7, Section 3.3: motivate why only maximum LAI is retuned for Sodankylä.
8) P.8, line 20: edit 'Given into account' to 'Taking into account'
9) P.9, lines 18-25: clarify the discussion. As the tuning aims at the 'best parameter', how could they be different?
10) The contents of Table 3 should be clarified, preferably in the bulk text where PCA is discussed. While the meaning of 'weight' is OK, the way the two most dominant parameters are calculated should be told to the reader.
11) Overall, the English language could be double-checked ('the' added in several places, etc)

A possible example reference for OAT methods:

Morris M (1991) Factorial sampling plans for preliminary computational experiments. Technometrics 33(2):161–174

---

## Editor Comment (EC1) · O. Talagrand (Editor) · 26 Sep 2016

Two referees have now sent their reports on the paper, and the open discussion on the manuscript has been closed. The authors have been given until October 15 to submit their response. I will then have to take my decision as editor.

In order to speed somewhat the whole process, I give here my comments and recommendations, based on the referees' reports and on my own opinion.

Referee 1 is an experienced specialist of land surface modelling, while Referee 2 is an expert on estimation theory. Both write that the paper contains new and significant results, but both also consider that a major revision is necessary before the paper can be accepted for publication. They make a fairly large number of comments and requests, which bear on the presentation of the work done by the authors, and on the conclusions to be drawn from the results they obtain. My understanding is that none of the referees' requests would require a significant change in the substance of the paper.

I follow the referees' advice, and suggest that the authors (if they have not already started doing so) prepare a revised version of their paper along the comments and requests of the referees.

The first request of Referee 1 is that the authors explain more precisely what is new and original in their paper. He/she also asks for a comparison of the model JSBACH with other ecosystem models, and questions the classification of the parameters in Table 1. He/she adds a number of specific comments.

Referee 2's comments bear more on the methodological aspects of the paper, and in particular on the use of the Adaptive Metropolis algorithm and the mathematical analysis of the results it produces. He/she also asks for a more detailed description of the JSBACH model (incidentally, it might be useful for some readers to specify clearly, as requested by the referee in his/her major comment 4, that JSBACH is a dynamical model which simulates the temporal evolution of an ecosystem). The referee adds a number of minor comments.

If, as I suggest them to do, the authors decide to prepare a revised version of their paper, they must do it in strict agreement with the instructions they have received from the Editorial Office of *Nonlinear Processes in Geophysics*. In particular, they must give a point-by-point response to all of both referees' comments and requests. Should they disagree with one particular comment, or decide not to follow one particular request, they must state precisely their reasons for that.

Concerning the last of Referee 2's requests (that the English be checked), I mention that, if the paper is accepted, it will in any case be checked for the English without additional cost for the authors.

The revised version will be submitted to further review by (normally) two referees who may, or may not, be the referees of the first version.

The authors must submit a response to the referees' reports by October 15. That response does not have to be their full revised version, which may come later if the authors need more time.

---

## Author Comment (AC1) · 14 Oct 2016

We thank both reviewers for their thorough and professional commenting on the manuscript. We have added a supplement "npg-response-files.zip" that includes the comments, updated version of the manuscript and the differences to previous version.

We answer the referee comments in the file "npg-responses-to-referees.pdf" where the comments have been numbered and indented. The other two files are "manuscript-Constraining_ecosystem_model-v2.pdf" as well as latexdiff output from the previous version named "manuscript-differences.pdf".

Please also note the supplement to this comment:

[Figure]

http://www.nonlin-processes-geophys-discuss.net/npg-2016-21/npg-2016-21-AC1-supplement.zip

---

## Author Response (AR1)

**Referee 1**

**General comments:**

- Parameters of the JSBACH land surface model are tuned for two forest sites in Finland. Photosynthesis and evapotranspiration estimates derived from eddy covariance measurements are used to calculate cost functions to be minimized. The optimization is able to correct for the main shortcomings of the model in the description of the annual cycle but is not sufficient to improve the representation of extreme events such as droughts. This shows that basic processes are missing in JSBACH. This kind of result is not new. The authors should do a better job in explaining what is new and original in their optimization approach. From a modelling perspective, a discussion is lacking about the reliability/robustness of JSBACH with respect to other models. From a methodology point of view, several issues need to be clarified. Spin-up must be performed for any new set of parameter values and it is not clear whether the authors made this effort or not. The purpose of the parameter classification (class I, II, and III) is not clear. The classification itself is not properly described, nor justified. Although the paper is reasonably well written, part of the method description is found in the Result section and should be moved to the Methods section. The Abstract need to be improved.

To our knowledge the Adaptive Metropolis algorithm has not been used in parameter sampling of LSMs. The reasons why we chose this algorithm were that it is robust in the terms of starting point and initial proposal covariance matrix of the parameters, even with multiple chains the use of this algorithm is straightforward and the use of multiple chains reduce the risk of the chains getting stuck.

We have added results from other models on these sites with further references and we have added section 2.3 "Model spin up and runs" to clarify the use of spin up. The use of single spin up is discussed at question 5.

The main purpose of the parameter classification was to reduce repetition – instead of writing LoGro phenology model parameters we can use Class III parameters. It has now been made clear in the text what are the distinctions of use between the different classes (no difference between II and III, and I is used only for the seasonal tuning). No methods should be found anymore in the Result section.

**Particular comments:**

1. P. 1, Abstract: A summary of the main findings regarding the usefulness of the optimization technique used in this study is lacking. Key results and conclusions must be listed.

We have added the requested findings in the abstract.

2. P. 3, L. 31: Why not including the spin-up into the calibration? Please clarify.

The purpose of the spin-up is to drive the model into a (semi)steady state at which point we have equilibrated the more slowly changing variables. During this process the variable values are unrealistic – for example LAI will take at least a decade to reach adequate levels, hence ET and GPP are also affected and should not be included in the metric. We have added a new subsection 2.3 "The JSBACH model spin up and runs" to clarify the use of the spin-up.

3. P. 4, L. 7: Some Class II and Class III parameters can also be "site-specific". For example, soil water retension parameters are highly site-specific. Please clarify what you mean by "site specific".

Our use of the term "site-specific" was taken from the point of view of a straightforward approach when making site simulations with a regional model. There the most effective parameters are optimised in order to improve the model performance at a site, neglecting the weak signals from other parameters. These dominating parameters may then incorrectly be called site specific, although the other parameters might also experience variability from site to site. In regional

modelling you anyway have to make compromises because of lack of data and let some of the parameters represent a larger region than their actual spatial variability allows. In JSBACH only one of the parameters examined (vegetative fraction of the grid cell) can vary site by site within a single run (so for regional runs all the parameters are the same). Since we also calibrate maximum LAI for the sites separately along with the carboxylation (and electron transport) rate for Hyytiälä, it seemed straightforward to use the term "site specific" for these parameters. We have now removed this ambiguous definition from the manuscript.

4. P. 5, L. 11-12: This argument is not valid as some Class II and Class III parameters listed in Table 1 can be site-specific. Do you mean that Class I parameters are observed and do not need any analysis ?

The reasoning to leave out Class I parameters from the analysis is that we consider the initial tuning as part of the model and experiment initialization. Hence the analysis of these parameters is not meaningful as they are used only to ensure a proper initial state for the daily and half-hourly tunings.

5. P. 6, L. 22: Is using a single spin-up valid ?

The single spin-up defines a reasonable initial state for the model since the robust initialization had already been done and the parameters in daily and half-hourly tunings affect the more "fine grained" processes (that also have a more immediate affect) in the model.

However we calculated the cost functions for tuned variables using this single spin-up and the reported values in Table 5 (where the spin-ups are generated using the tuned values) and the differences in the cost functions are less than 1 % (daily) and less than 0.1 % (half-hourly). Approximately 6 % of parameters tested in the MCMC process yield a cost function value below a corresponding threshold for daily tuning and significantly less than 0.1 % for half-hourly tuning. With this we would claim that the approach is valid for our experiments although this claim should not be generalized.

6. P. 7, L. 8: Does this mean that class I parameters other than maximum LAI are not considered as site-specific ?

This question has been touched above as we discussed the term "site specific". In this study only maximum LAI (of the given parameters) differs between the two sites. This claim holds also for Class II and III parameters.

7. P. 9, L. 4: This paragraph is difficult to understand because the methods were not sufficiently described and symbols were not defined before. Methods, as well as "L1", and all the other symbols of Table 5 (including "HC", "HV", "SV") should be defined/presented in Sect. 2. Not here in the result Section

We have now replaced ∑ signs with corresponding cost function abbreviations in Table 5, defined these and "L1,E1,G1" within the cost function definitions. Additionally abbreviations "HC", "HV" and "SV" are now define in subsection 2.3.

8. P. 9, L. 6 ("half as large"): Half as large as what ?

We have now amended the sentence [additions]: "As expected the L1 for Sodankylä is not as dominant as for Hyytiälä since the measured maximum of LAI [for Hyytiälä] is roughly half as large [as for Sodankylä], which directly lowers the LAI component in cost function (1)."

9. P. 9, L. 14 ("ET is a more turbulent flux than GPP"): What do you mean ? GPP is not a turbulent flux at all. The turbulent $CO_2$ flux is NEE, not GPP. GPP is not directly measured by eddy covariance techniques.

This is absolutely true. What we were trying to say (briefly) is that the time series for ET is much more erratic in comparison to GPP and the residuals of observed and (JSBACH) modelled GPP are smaller in comparison to ET (as we also divide the residuals with the mean of observed values in cost function 2). This sentence has now been amended.

10. P.9, L. 27: The JSBACH model simulations don't look very good. How does JSBACH perform with respect to other models at these two sites ? Please give basic scores in terms of half-hourly fluxes, such as RMSD, ubRMSD and mean bias.

We have added RMSE and bias estimates of the given time series to Table 6 and compare these to PRELES model (unfortunately no RMSE/RMSD type of estimates are given for PRELES).

11. P. 11, L. 1: How can this be explained ? Shortcomings in the representation of the soil moisture stress ? How could these shortcomings be attenuated ? Using another photosynthesis model ?

The shortcomings are rather attributed to (Gao et al 2016) the lack of explicit dependence of stomatal conductance to air humidity that leads to deviating behavior between model and observations under severe soil moisture stress. The shortcomings can be attenuated by implementing explicit dependence of conductance on VPD. This may require selection of different formulation of photosynthesis model.

**Editorial comments:**

1. P. 18 (Table 1): Parameters' units are lacking.

Units have been added to Table 1.

2. P. 19 (Table 4): Parameters' units are lacking.

Units have been added to Table 4.

3. P. 20 (Table 5, "highlighted values"): I don't see any highlighted value.

This table was previously in another form and the mention of the highlighted values is redundant. We have also amended Table 5 and removed the mention of highlighted values.

**Referee 2**

**Major Comments**

1) To estimate the distribution of parameters B of a model F based on data Y given by experiments X, connected by the standard expression 'Y= F(X,B) + eps', the distribution of the measurement error 'eps' should be known. But here the authors give almost no information of any of these to a reader not already familiar with JSBACH and the measurements. Certainly it is not possible to give all details, but the basic parts of the underlying modeling and numerical solution should be described, maybe in an Appendix, not to leave F(X,B) as a fully black box for the reader. See comments 4), 6) and 7) below.

We have now added a description about the measurement errors to the manuscript (at the end of section 2.1 Measurements, sites and instrumentation) and reference to the MPI-ESM model description (which includes JSBACH). The main equation have also been added to "Appendix A: Parametric equations within JSBACH".

2) To optimize the model parameters the Adaptive Metropolis (AM) method is chosen. It is, however, a sampling method rather than optimization. The motivation and benefits of the choice should be given: instead of a point estimate, samples of the full distribution of possible parameter values are obtained, together with (nonlinear) correlation information, sensitivity, identifiability of parameters, etc.

We have complemented and expanded our description of the AM method in chapter 2.5 "Parameter sampling".

3) The parameter estimation is based on the two cost functions on p. 4 and 5. But no info is given here on the assumed statistics of the expressions, only a hint on Gaussian distribution later on p. 7. Usually, the sum of squares of the residuals is divided by the respective estimated variance of measurement error. Here, the residuals are normalized by the observations. This can be quite acceptable if no 'true' error statistics is available, and the

sampling is done in the spirit of studying the identifiability and correlations of the parameters. However, this should be done explicit in the text.

We have added coupling of likelihood function and cost functions to the article, as well as description about measurement errors. We have also added our motivation for normalizing the sums with a mean of observations (we have only a general type of error for the point estimates).

4) For the general audience (not familiar with JSBACH) at least the basics of the numerical approach used in JSBACH should be given, together with the CPU demands of the runs. Now only an implicit statement (' ...interval is looped over to generate a 30 year spin up ...', line 30 , p.3) is given that would indicate that JSBACH is a dynamic model that has to be initialized or run into a (quasi) steady-state to compare with observations ? Or is this due to the uncoupled version used here? The concept and use of spin-up should be clarified.

The CPU demands have now been added to the start of section 3 "Model tuning". The JSBACH itself is roughly 100 000 lines of code (in Fortran). In approach it is an process based model so the processes in JSBACH mimic those in nature e.g. differential equations for heat diffusion in soil. In solving these, various methods are used, such as replacing nonlinear terms with truncated Taylor expansions. We have now included a reference to Echam (atmospheric component of MPI-ESM) model description which includes JSBACH. We have also added section 2.3 "The JSBACH model spin up and runs" to clarify the use of the spin-up (to equilibrate e.g. LAI and as suggested above to bring the model into a steady state).

5) How much does the uncoupling impact the results in general? The authors mention (P.10, line 13-15) that the lack of coupling of the LSM model to atmosphere generates an erroneous energy balance. This aspect should be discussed or commented more explicitly.

We have now briefly discussed the uncoupling in the beginning of section "2.2 JSBACH model". This question is not a trivial one and could actually be a topic for another (couple) of papers. In our simulations nighttime and wintertime negative evapotranspiration values are attributed to surface temperatures that are slightly lower than air temperatures from the meteorological drivers. This, accompanied with turbulent mixing that is driven with prescribed wind speed and obviously not suppressed enough under these stable stratification situations maintain condensation at the surface throughout periods that lack diabatic heating by the shortwave radiation from the sun. Holtslag et al. 2007 (http://edepot.wur.nl/37199) have emphasized the importance of the mutual consistence among the drivers regulating temperature and momentum in order to achieve realistic magnitudes of turbulent fluxes under stable conditions.

6) The discussion in Section 2.5, parameter posterior distribution vs PCA, is not clear. The authors 'perform a PCA analysis transforms of the covariance matrices ...' – but do not tell what covariance ? My guess would be that they actually mean the matrix of the AM samples of parameter vectors, and compute the PCA of it to get the eigenvectors of the least identified parameter directions. This can lead to correct conclusions, assuming that the nonlinear correlations between the parameters are not too strong. That, on the other hand, is typically indicated by plotting the 2D scatter plots of parameter marginal distributions. So I would recommend the authors to show them as well, and clarify the discussion on how PCA was used.

Originally we meant a covariance matrix derived from the tested parameter samples, which was then divided by the root of the product of variances (which does produce the correlation matrix). We have now revised this section and omitted the mention of "covariance" in favor of the correlation (since this could also be nuisance to readers unfamiliar with the method). We have also added kernel density estimates instead of the different parameters (we tried the 2D scatter plots but it was difficult to get any information from these visually).

7) No information is given on how the studied parameters appear in the model. It is wellknown that the parametrizations strongly impact the identifiability. A good example is the logistic function, where centering and scaling typically removes correlations. So this point should be made explicit by showing the formulas, at least in case of the LoGro phenology model where high correlations appear 'since the parameters are intimately connected' (L.30, p.7).

We have now added "Appendix A: Parametric equations within JSBACH" that give the main equations for all parameters examined.

8) The measurements consist of the CO2 fluxes as given by the eddy covariance method. But the cost functions are given in terms of 'observed' and modeled GPP,ET and LAI. The connection between CO2 fluxes and those cost function expressions should be given.

This connection has now been added to section 2.1 "Measurements, sites and instrumentation".

**Minor comments**

1. In addition to the PCA/MCMC analysis of least identified parameters, the authors study which parameters are the most relevant for the change of the cost function. They introduce an OAT (one-at-a-time) method of their own (?). The relation of it to well-known methods such as the MOAT (Morris-OAT, see the reference below) could be make more clear. Also, it is not clear what the 'tuned parameter' (p.5, Step 1) is: the mean of the sampled values, or the maximum likelihood (minimum cost function) value? I would gather that the 'reference value' is the initial/default value of optimization. These points should be made clear.

We have now added a more thorough description about our OAT method. The definition of tuned parameters has been added to the start of section "Parameter analysis".

2. Only the cost functions are given in the text, not the likelihood used in the sampling. If it is Gaussian as indicated on p.7, it should be mentioned that the 'f' function of step 2., p. 4, actually is the exponential function of the (negative) cost function.

These clarifications have now been added to the manuscript.

3. P.4 line 15: The sentence 'A sample in the parameter has a value ...' could be removed. Instead, the term 'chain' could be explained for a reader unfamiliar with MCMC.

The sentence mentioned has been removed and we have added a short description of the MCMC chain.

4. P.4, line 16: edit the sentence 'The algorithm is used ...' something like 'The algorithm can be used' or 'is used here', since the basic form of the AM algorithm is or a single chain. Maybe add a reference to parallel chain adaptive MCMC

This has been amended and we have added two references for parallel chain adaptive MCMC.

5. P. 6, line 20/Step 3: 'Initial covariance' means the initial proposal covariance for MCMC sampling ?

Yes – added "proposal" to text.

6. P. 6, lines 17 and 27: it would be good to know here how many parameters were used for the 10000 sample long chains.

The number of parameters has now been added.

7. P. 7, Section 3.3: motivate why only maximum LAI is retuned for Sodankylä.

We have now added our motivation to use Sodankylä as a validation site to optimization done with another boreal forest site.

8. P.8, line 20: edit 'Given into account' to 'Taking into account'

Amended.

9. P.9, lines 18-25: clarify the discussion. As the tuning aims at the 'best parameter',

how could they be different?

They should not be different. This part of the discussion was to point out that we have not made any gross mistakes/violations is the tuning.

10. The contents of Table 3 should be clarified, preferably in the bulk text where PCA is discussed. While the meaning of 'weight' is OK, the way the two most dominant parameters are calculated should be told to the reader.

The basis of calculations has been added to the bulk of text where PCA is discussed.

11. Overall, the English language could be double-checked ('the' added in several places, etc)

As stated by the editor, the language will be checked prior to publishing, if the manuscript is accepted. Although we have made some efforts to recheck the language.

[revised manuscript text omitted]

---

## Referee Report (RR1)

2.nd Report on the article 'Makela,
Susiluoto,Markkanen,Aurela,Mammarella,Hagemann, Aalto: Constraining ecosystem
model with Adaptive Metropolis algorithm using boreal forest site eddy covariance
measurements'

As stated in my first review, the work is certainly professional and worthwhile to
publish. The critical comments have now been take into account, and the presentation
has improved. Naturally, smaller items would remain, but as for me the present
version is good enough to be published. I understand that the editors  & authors take
are of the language yet.

Some Minor Comments yet:

1) Remove the copy-paste typo, rows 11-18, page 4
2) Check style for repetitions, e.g,  'presentation of the Am algorithm is
   presented' , row 7, p. 6
3) 'least concergent' → 'least identified', row 26, p. 7
4) 'change is value' → 'change in value', row 17, p. 8

---

## Editor Decision (ED1)

Dear Dr. Mäkelä,

I have now received two reports of the revised version of your paper. The referees are the same as those of the first version, with the same identification numbers.

Both referees consider your paper can now be accepted for publication. Referee 2 mentions a few editing corrections (the line and page numbers that the referee mentions for those corrections do not correspond to either one of the two versions of your revised paper I have, but I think you will have no difficulty in finding where to make the corrections).

I add as editor a few additional corrections that I think desirable.

1. P. 7, ll. 12-13, *In a case that the parameter chains converge to a limit of a predescribed range* … From what I understand, … *converge to a bound of an a prori prescribed range* … would be preferable.

2.  Abstract, l. 6, … *possible parameters,* …

3.  Abstract, l. 12. Expand explicitly the acronyms GPP and ET.

4.    Caption of Figure 1. *The contours correspond to* […] *normal distribution*. Do you mean *normalized* (but not Gaussian !) ?

5. P. 8, l. 18  (and p. 12, l. 27), what is the *coefficient of determination* (the correlation coefficient ) ?

6. P. 7, l. 22, … *the squared weights sum up to one*

Please make the corrections suggested by referee 2 as well as by myself. If you decide not to make a particular correction, say why.

I thank you for having chosen *Nonlinear Processes in Geophysics* for publishing your paper, and look forward to receiving the final version.

Olivier Talagrand

Editor, *Nonlinear Processes in Geophysics*

---

## Author Response (AR2)

**Responses to referees and the editor**

We thank again both referees and the editor for providing valuable feedback to our manuscript. All requested changes have been made to the manuscript. There were however two comments from the editor we felt should be addressed. Additionally we have made few other minor corrections to the manuscript.

**Editorial comments**

4. Caption of Figure 1. The contours correspond to [...] normal distribution. Do you mean normalized (but not Gaussian !) ?

We mean two dimensional gaussian distributions. The density within a given contour corresponds to a density within the corresponding contour of a two dimensional gaussian distribution (so the integral over this contour is the same for both).

5. P. 8, l. 18 (and p. 12, l. 27), what is the coefficient of determination (the correlation coefficient ) ?

The equation for calculating the coefficient of determination has been added to page 8. This is generally not the same as the correlation coefficient (as there are multiple definitions for it).

[revised manuscript text omitted]